# FEDERATED DOMAIN GENERALIZATION WITH DATA-FREE ON-SERVER MATCHING GRADIENT

**Trong-Binh Nguyen**[1*]**, Minh-Duong Nguyen**[1*]**, Jinsun Park**[1†]**, Quoc-Viet Pham**[2]**, Won Joo Hwang**[1†]
[1] Pusan National University, Republic of Korea, [2] Trinity College Dublin, Ireland
[*] Equal contribution, [†] Corresponding author
{binhnguyentrong, duongnm, jspark, hwangwj}@pusan.ac.kr
viet.pham@tcd.ie

## ABSTRACT

Domain Generalization (DG) aims to learn from multiple known source domains a model that can generalize well to unknown target domains. One of the key approaches in DG is training an encoder which generates domain-invariant representations. However, this approach is not applicable in Federated Domain Generalization (FDG), where data from various domains are distributed across different clients. In this paper, we introduce a novel approach, dubbed Federated Learning via On-server Matching Gradient (FedOMG), which can *efficiently leverage domain information from distributed domains*. Specifically, we utilize the local gradients as information about the distributed models to find an invariant gradient direction across all domains through gradient inner product maximization. The advantages are two-fold: 1) FedOMG can aggregate the characteristics of distributed models on the centralized server without incurring any additional communication cost, and 2) FedOMG is orthogonal to many existing FL/FDG methods, allowing for additional performance improvements by being seamlessly integrated with them. Extensive experimental evaluations on various settings demonstrate the robustness of FedOMG compared to other FL/FDG baselines. Our method outperforms recent SOTA baselines on four FL benchmark datasets (MNIST, EMNIST, CIFAR-10, and CIFAR-100), and three FDG benchmark datasets (PACS, VLCS, and OfficeHome). The reproducible code is publicly available [1].

## 1 INTRODUCTION

Federated Learning (FL) has gained widespread recognition due to its ability to train models collaboratively across multiple clients while keeping their individual data secure. However, one practical challenge, how to ensure that models trained on sites with heterogeneous distributions generalize to target clients with unknown distributions, known as Federated Domain Generalization (FDG), remains under-explored. While label distribution shift has been considered in traditional FL, FDG focuses on the feature shift among clients and considers each client as an individual domain.

FDG shares a similar goal as standard Domain Generalization (DG) (Nguyen et al., 2022b; Li et al., 2018), i.e., generalizing from multi-source domains to unseen domains. However, unlike DG, where knowledge from various domains can be jointly utilized to develop an efficient algorithm, FDG prohibits direct data sharing among clients, which makes most existing DG methods hardly applicable. To address the challenges inherent to FDG, recent methods circumvent these difficulties by adopting alternative strategies, e.g., unbiased local training within each isolated domain, local data preprocessing (Huang et al., 2022; 2024), or knowledge sharing among users (Liu et al., 2021; Chen et al., 2023). As a consequence, an open question remains for FDG:

*How can knowledge across domains be effectively leveraged to design FDG algorithms that achieve performance comparable to DG, while avoiding additional communication overhead?*

---

[1] https://github.com/skydvn/fedomg

Addressing the aforementioned question, we inherit the gradient matching rationale (Shi et al., 2022; Rame et al., 2022). The rationale is straightforward: if the model $\theta$ is domain-invariant, the gradients induced by $\theta$ across different domains should exhibit a high correlation with each other. To this end, we propose a novel on-server invariant gradient aggregation approach for FDG, dubbed Federated learning via On-server Matching Gradient (FedOMG). Specifically, we leverage the available local gradients at the server to design a gradient-based approach for finding domain-invariant models. To design an on-server optimization strategy, we draw inspiration from the Gradient Inner Product (GIP) optimization problem introduced in Fish (Shi et al., 2022) to implement gradient matching. However, the current GIP approach has two significant limitations for FDG: (1) directly minimizing GIP, as described in (Shi et al., 2022, Alg. 2), incurs substantial computational overhead due to the need to compute second-order derivatives of model parameters related to the GIP term; and (2) the surrogate approach for Fish, as outlined in (Shi et al., 2022, Alg. 1), requires continuous transmission of client models, leading to excessive communication overhead. To address these issues, we propose an indirect optimization method that leverages surrogate optimization variables instead of model parameters, thereby eliminating the need for second-order derivatives in GIP. Furthermore, to enhance computational efficiency, we introduce an efficient convex optimization formulation to streamline the on-server optimization process. The advantages of our design are as follows:

1. FedOMG can leverage local client knowledge to train a global model, similar to conventional DG. Consequently, our global model has a better knowledge of all clients, thus, leading to high generalization when applied to Out-Of-Distribution (OOD) data.

2. Due to the design of on-server gradient matching, our method is orthogonal to many classic FDG methods, where current FDG methods focus on the local training (Huang et al., 2022; 2024). As a consequence, the integration of FedOMG with other FDG methods can achieve improved performance.

3. Due to the utilization of available local gradients on server, our method does not require data transmission from distributed devices to the server, or sharing data among users (Liu et al., 2021; Chen et al., 2023). Consequently, we can maintain the efficiency of communication and privacy in FL settings.

**Contribution.** Our contributions can be summarized as follows. 1) We propose FedOMG, a communication-efficient method for learning domain-invariant models in FDG, which leverages the local gradients to facilitate the on-server gradient matching. 2) We conduct theoretical analysis to reveal that FedOMG is robust to the reduction of the generalization gap of the unseen target domains. 3) We conduct experimental evaluations under FL settings and FDG settings. In FL settings, we consider four FL datasets (MNIST, EMNIST, CIFAR-10, and CIFAR-100). In FDG settings, we consider the DG benchmark datasets (PACS, VLCS, and OfficeHome). Experimental results show that FedOMG significantly outperforms existing FL methods in terms of IID, non-IID dataset, and OOD generalization.

## 2 PRELIMINARIES

**Notations:** By default, $\langle \cdot, \cdot \rangle$ and $\| \cdot \|$ denote the Euclidean inner product and Euclidean norm, respectively. In our research, we use $d_{\mathcal{M}}(\cdot, \cdot)$ to represent any distance metric $\mathcal{M}$ between two distributions. We denote $\odot$ as Hadamard product. We denote $\mathbb{P}(\theta; \mathcal{D})$ as the output distribution of model $\theta$ given input as dataset $\mathcal{D}$. We denote $\theta_g^{(r)}, \theta_u^{(r)}, \phi_u^{(r)}, \in \mathbb{R}^{1 \times M}$ as the global, local, trained local model parameters used in the FL system at round $r$, respectively. Let $g(\theta_u^{(r,e)}; \mathcal{D}_u) \in \mathbb{R}^{1 \times M}$ be local gradient of client $u$ at epoch $e$ of round $r$, $g(\theta^{(r)}; \mathcal{D}_u) \in \mathbb{R}^{1 \times M}$ be gradient of client $u$ at round $r$. Without loss of generality, we abuse $g(\theta_u^{(r,e)}; \mathcal{D}_u) = g_u^{(r,e)}$, $g(\theta^{(r)}; \mathcal{D}_u) = g_u^{(r)}$.

**Problem Setup:** We consider an FL system comprising a set of source clients $\mathcal{U}_{\mathcal{S}} = \{u | u = 1, 2, \ldots, U_{\mathcal{S}}\}$, and target clients $\mathcal{U}_{\mathcal{T}} = \{u | u = U_{\mathcal{S}} + 1, U_{\mathcal{S}} + 2, \ldots, U_{\mathcal{S}} + U_{\mathcal{T}}\}$. Each client $u$ gains access to its data $\mathcal{D}_u = \{x_i, y_i\}_{i=1}^{N_u}$ with $N_u$ data samples, where $u \in \{\mathcal{U}_{\mathcal{S}}; \mathcal{U}_{\mathcal{T}}\}$. Source and target clients are assigned to the source datasets $\mathcal{D}_{\mathcal{S}} = \{\mathcal{D}_u | u \in \mathcal{U}_{\mathcal{S}}\}$ and target datasets $\mathcal{D}_{\mathcal{T}} = \{\mathcal{D}_u | u \in \mathcal{U}_{\mathcal{T}}\}$, respectively. Each client is trained for $E$ epochs every round. The system objectives are two-fold.

**Objective 1 (guarantee on the conventional FL setting)** *The method has to perform well on the source clients set $\mathcal{U}_S$, i.e., $\theta^* = \arg\min_\theta \mathcal{E}_S = \sum_{u \in \mathcal{U}_S} \gamma_u \mathcal{E}(\theta, \mathcal{D}_u)$, where $\mathcal{E}(\theta, \mathcal{D}_u)$ is the local empirical risk, $\gamma_u$ represents the weight on client $u$, satisfying $\sum_{u \in \mathcal{U}_S} \gamma_u = 1$.*

**Objective 2 (guarantee on the FDG setting (Zhang et al., 2023b))** *Let $\mathcal{H}$ be a hypothesis space of VC-dimension $M$, $d_{\mathcal{H} \triangle \mathcal{H}}(\mathcal{D}_u, \mathcal{D}_v)$ is the domain divergence. For any $\delta \in (0, 1)$, the generalization gap on an unseen domain $\mathcal{D}_T$ is bounded by the following with a probability of at least $1 - \delta$:*

$$\mathcal{E}(\theta; \mathcal{D}_T) \leq \sum_{u \in \mathcal{U}_S} \gamma_u \left[ \mathcal{E}(\theta; \mathcal{D}_u) + \sum_{v \in \mathcal{U}_T} d_{\mathcal{H} \triangle \mathcal{H}}(\mathcal{D}_u, \mathcal{D}_v) + \sqrt{\frac{\log M + \log \frac{1}{\delta}}{2 N_u}} \right] + \zeta^*, \quad (1)$$

*where $\zeta^*$ is the optimal combined risk on $\mathcal{D}_S$ and $\mathcal{D}_T$. One crucial challenge in FDG is the existence of a domain shift between the sets of source clients' data $\mathcal{D}_S$ and target clients' data $\mathcal{D}_T$, respectively.*

Objectives 1 and 2 highlight two primary goals of FDG. While Objective 1 can easily be achieved by training on source clients $\mathcal{U}_S$, Objective 2 requires the FDG system to minimize the discrepancy between data from source clients $\mathcal{U}_S$ and unseen target clients $\mathcal{U}_T$. However, Inequality 1 requires the availability of both source and target data (i.e., $\mathcal{D}_u \in \mathcal{D}_S, \mathcal{D}_v \in \mathcal{D}_T, \forall u, v \in \mathcal{U}_S, \mathcal{U}_T$). Consequently, one of the most significant approaches to achieving generalization is to design a model $\theta$ that is domain-invariant (Li et al., 2018; Zhao et al., 2019) by learning through the objective:

$$\theta^* = \arg\min_\theta \mathcal{E}[\theta; \mathcal{D}_S] + \lambda \sum_{u \in \mathcal{U}_S} \sum_{v \in \mathcal{U}_S}^{v \neq u} d_{\mathcal{M}}(\mathbb{P}(\theta; \mathcal{D}_u), \mathbb{P}(\theta; \mathcal{D}_v)), \quad (2)$$

where $\lambda$ is the scaling coefficient of domain-invariant regularization. The principle of domain-invariant learning is to learn a model that remains invariant across source domains, rather than reducing the divergence between the source and unavailable target domains. As a consequence, domain-invariant learning efficiently reduces the generalization gap between source and target domains (Li et al., 2018). However, due to the decentralization of the data, each client does not have access to other clients' data, i.e., $\mathcal{D}_u \cap \mathcal{D}_v = \varnothing, \forall u, v \in \mathcal{U}_S, u \neq v$. Consequently, it is challenging to find the domain-invariant features within the FDG framework.

## 3 MOTIVATION BEHIND ON-SERVER OPTIMIZATION FEDERATED LEARNING

### 3.1 FROM FEDERATED LEARNING TO ON-SERVER OPTIMIZATION FEDERATED LEARNING

To efficiently aggregate knowledge from all clients, we propose an on-server optimization framework designed to identify an optimal aggregated gradient for achieving domain invariance, thus, eliminating the need for direct access to client data. This approach leverages meta-learning to decompose the FDG optimization problem into two meta-learning steps. Specifically, we reformulate the FDG optimization problem from Eq. (2) into two sequential stages: local update and meta update. Drawing on the meta-learning principle (Hospedales et al., 2022), the problem is formulated as follows:

$$\theta_g^{(r)} = \theta_g^{(r-1)} - \eta_g \sum_{u \in \mathcal{U}_S} \left[ \nabla \mathcal{E}(\phi_u^{(r)}; \mathcal{D}_u) + \lambda \sum_{v \in \mathcal{U}_S}^{v \neq u} \nabla d_{\mathcal{M}}(\mathbb{P}(\theta; \mathcal{D}_u), \mathbb{P}(\theta; \mathcal{D}_v)) \right], \quad (3a)$$

$$\text{s.t.} \quad \phi_u^{(r)} = \theta_g^{(r-1)} - \eta_l g(\theta_g^{(r-1)}; \mathcal{D}_u), \quad (3b)$$

where $\eta_l$ and $\eta_g$ are the local and global learning rate of FL system, respectively. The local update in Eq. (3b) is designated for on-device training, while the meta update in Eq. (3a) is utilized for on-server update. The on-server update in (3a) consists of two terms: the first term $\nabla \mathcal{E}(\phi_u^{(r)}; \mathcal{D}_u)$ refers to the FL update, and the second term $\nabla d_{\mathcal{M}}(\mathbb{P}(\theta; \mathcal{D}_u), \mathbb{P}(\theta; \mathcal{D}_v))$ represents the update of domain divergence reduction between any pair of clients from the source client set $\mathcal{U}_S$. However, as aforementioned in Section 2, the integration of Eq. (2) into Eq. (3) is infeasible due to the requirement of accessibility to all source clients' data.

### 3.2 Motivation of Inter-domain Gradient Matching

Addressing the demand for local data access in conventional DG approach, we utilize *Invariant Gradient Direction* (IGD):

**Definition 1 (Invariant Gradient Direction (Shi et al., 2022))** *Considering a model $\theta$ with task of finding domain-invariant features, the features generated by the model $\theta$ is domain-invariant if the two gradients $g(\theta; \mathcal{D}_u)$, and $g(\theta; \mathcal{D}_v)$ point to a similar direction, i.e., $g(\theta; \mathcal{D}_u) \cdot g(\theta; \mathcal{D}_v) > 0$. Otherwise, the invariance cannot be guaranteed if $g(\theta; \mathcal{D}_u) \cdot g(\theta; \mathcal{D}_v) \leq 0$.*

Given Definition 1, we can leverage the local gradients as data for on-server optimization. Recently, many researches have been carried out to find the invariant gradient direction among domains. However, most of the researches consider gradient minimization as a regularization of the joint objective function. For instance, Fishr (Rame et al., 2022, Eq. (4)) considers

$$\mathcal{E}_{\text{Fishr}} = \frac{1}{U_{\mathcal{S}}} \sum_{u \in \mathcal{U}_{\mathcal{S}}} \mathcal{E}(\theta^{(r)}; \mathcal{D}_u) + \lambda \frac{1}{U_{\mathcal{S}}} \sum_{u \in \mathcal{U}_{\mathcal{S}}} \|a_u^{(r)} - a^{(r)}\|^2, \text{ s.t. } a_u^{(r)} = \frac{1}{N_u} \sum_{i=1}^{N_u} (g_{u,i}^{(r)} - g_u^{(r)})^2, \quad (4)$$

where $g_u^{(r)} = \frac{1}{N_u} \sum_{i=1}^{N_u} g_{u,i}^{(r)}$, and $a^{(r)} = \frac{1}{U_{\mathcal{S}}} \sum_{u \in \mathcal{U}_{\mathcal{S}}} a_u^{(r)}$ are the client mean gradient and global mean gradient variance, respectively. Another approach is to use GIP as regularization. For instance, Fish (Shi et al., 2022, Eq. (4)) considers

$$\mathcal{E}_{\text{Fish}} = \frac{1}{U_{\mathcal{S}}} \sum_{u \in \mathcal{U}_{\mathcal{S}}} \mathcal{E}(\theta^{(r)}; \mathcal{D}_u) - \lambda \frac{2}{U_{\mathcal{S}}(U_{\mathcal{S}} - 1)} \sum_{u \in \mathcal{U}_{\mathcal{S}}} \sum_{v \in \mathcal{U}_{\mathcal{S}}}^{v \neq u} \left\langle g_u^{(r)}, g_v^{(r)} \right\rangle. \quad (5)$$

On-server data are required if we leverage Eq. (4) or Eq. (5) as the objective function for on-server optimization. Acknowledge this shortcoming, we design a *meta-learning based approach* and apply the gradient matching solely during the meta-update stage. Therefore, the FDG update can be represented as follows:

$$\theta_g^{(r)} = \theta_g^{(r-1)} - \eta_g g(\theta_{\text{IGD}}^{(r)}; \cdot), \quad (6a)$$

$$\text{s.t.} \quad \theta_{\text{IGD}}^{(r)} = \arg\max_{\theta} \sum_{u \in \mathcal{U}_{\mathcal{S}}} \left\langle g(\theta; \cdot), g(\phi_u^{(r)}; \mathcal{D}_u) \right\rangle, \quad \phi_u^{(r)} = \theta_g^{(r-1)} - \eta_l g(\theta_g^{(r-1)}; \mathcal{D}_u). \quad (6b)$$

Here, we abuse $g_u^{(r)} = g(\phi_u^{(r)}; \mathcal{D}_u)$ as the local gradient of user $u$ on round $r$ after $E$ epochs. $g(\theta; \cdot) = \theta - \theta_g^{(r-1)}$ is the gradient computed by learnable parameter $\theta$, thus, we do not consider the dataset. Note that Eq. (6) has a high similarity with (Shi et al., 2022, Eq. (4)). Apart from (Shi et al., 2022), we use $\sum_{u \in \mathcal{U}_{\mathcal{S}}} \left\langle g(\theta; \cdot), g(\phi_u^{(r)}; \cdot) \right\rangle$ instead of $\sum_{u \in \mathcal{U}_{\mathcal{S}}} \sum_{v \in \mathcal{U}_{\mathcal{S}}} \left\langle g_u^{(r)}, g_v^{(r)} \right\rangle$ because $\langle g_u^{(r)}, g_v^{(r)} \rangle$ is double-edged bounded by $\langle g(\theta; \cdot), g_v^{(r)} \rangle$ and $\langle g(\theta; \cdot), g_u^{(r)} \rangle$, $\forall u, v$, as explained in the following.

**Lemma 1 (Triangle inequality for cosine similarity (Schubert, 2021))** *Let $g_u^{(r)}, g_v^{(r)}, g(\theta; \cdot)$ be three vectors in a $M$-hyperplane, then the following bounds hold:*

$$\langle g_u^{(r)}, g(\theta; \cdot) \rangle \langle g(\theta; \cdot), g_v^{(r)} \rangle - \sqrt{(1 - \langle g_u^{(r)}, g(\theta; \cdot) \rangle^2)(1 - \langle g(\theta; \cdot), g_v^{(r)} \rangle^2)} \leq \langle g_u^{(r)}, g_v^{(r)} \rangle$$

$$\leq \langle g_u^{(r)}, g(\theta; \cdot) \rangle \langle g(\theta; \cdot), g_v^{(r)} \rangle + \sqrt{(1 - \langle g_u^{(r)}, g(\theta; \cdot) \rangle^2)(1 - \langle g(\theta; \cdot), g_v^{(r)} \rangle^2)}. \quad (7)$$

By proposing Eq. (6), we establish the meta objective function Eq. (6a), which leverages only the clients' gradient as training data. The remaining issues of Eq. (6) are two-folds. Firstly, the optimization problem over the model $\theta$ may not achieve good generalization due to the *lack of training data* at every optimization step (which is not applicable to model with large parameters). Secondly, as noted by Shi et al. (2022), minimizing the GIP can result in *significantly high computational overhead*, primarily due to the requirement of computing second-order derivatives of the model parameters associated with the GIP term. In our proposed method, we will focus on making this objective function feasible while not require extensive computational resources due to the second-order derivatives of the model parameters.

## 4 FedOMG: Data Free On-Server Matching Gradient

### 4.1 Overall System Architecture

We propose a novel method FedOMG, which efficiently aggregates the clients' knowledge to learn a generalized global model $\theta_g$. The main differences between our approach and other current FL and FDG approaches are two-fold. Firstly, to guarantee the communication efficiency of FL settings, we leverage the available *local gradients as training data* for our on-server optimization.

$$g_u^{(r)} = \theta_u^{(r,E)} - \theta_u^{(r,0)}, \quad \text{where} \quad \theta_u^{(r,0)} = \theta_g^{(r-1)}. \tag{8}$$

Secondly, we design a novel *on-server optimization* approach. The goal of this on-server optimization is to find an encoder with the capability of generating domain-invariant features. To this end, we leverage clients' gradients in the global model $\theta_g$ that achieves the invariant gradient direction on all domains.

### 4.2 Inter-client Gradient Matching

To make the objective function (6) feasible, we target to do the following: 1) limit the searching space of the meta-update, 2) leverage Pareto optimality to reduce the computation overheads, and 3) indirectly find invariant gradient direction.

**Indirect Search of Invariant Gradient Direction.** As mentioned in Section 3.2, the utilization of GIP may induce significant computation overheads due to the on-server training, as the optimization over model parameter $\theta$ requires the second order derivative. To this end, instead of finding a gradient solution with $M$-dimensional optimization variable $\theta_{\text{IGD}}^{(r)} = \arg\max_\theta \sum_{u \in \mathcal{U}_S} \langle g(\theta; \cdot), g(\phi_u^{(r)}; \cdot) \rangle$, we consider the invariant gradient direction $g_{\text{IGD}}^{(r)}$ as a convex combination of local gradients $g_u^{(r)}$.

$$g_{\text{IGD}}^{(r)} = \Gamma \mathbf{g}^{(r)} = \sum_{u \in U_S} \gamma_u g_u^{(r)}, \text{ where, } \Gamma = \{\gamma_1, \ldots, \gamma_{U_S}\}, \ \mathbf{g}^{(r)} = \{g_1^{(r)}, \ldots, g_{U_S}^{(r)}\}. \tag{9}$$

By doing so, the FDG update in Eq. (6) is reduced to

$$\theta_g^{(r)} = \theta_g^{(r-1)} - \eta_g g_{\text{IGD}}^{(r)} = \theta_g^{(r-1)} - \eta_g \Gamma_{\text{IGD}} \mathbf{g}^{(r)}, \tag{10a}$$

$$\text{s.t.} \quad \Gamma_{\text{IGD}} = \arg\max_\Gamma \sum_{u \in \mathcal{U}_S} \left\langle \Gamma \mathbf{g}^{(r)}, g(\phi_u^{(r)}; \mathcal{D}_u) \right\rangle, \quad \phi_u^{(r)} = \theta_g^{(r-1)} - \eta_l g(\theta_u^{(r,E)}; \mathcal{D}_u). \tag{10b}$$

By indirectly optimizing the joint gradients over the auxiliary optimization set $\Gamma$, the need for computing second-order derivatives with respect to the model parameters $\theta$ is eliminated. Additionally, the optimization variable dimensionality is reduced from $M$ to $U_S$ ($M \gg U_S$), which further improve the computational efficiency.

**Searching Space Limitation.** The straightforward optimization of GIP, as formulated in Eq. (10), may introduce an optimization bias toward gradients with the dominating magnitudes. This bias can result in a loss of generalization, potentially causing the optimization process to overlook clients that contribute less significantly during a given communication round. An alternative approach for GIP involves leveraging cosine similarity. By doing so, the influence of gradient magnitudes among local gradients is minimized, allowing the focus to shift to optimizing the angles between gradients. However, this method is computationally intensive and challenging to simplify into a more practical formulation. To this end, we limit the searching space into the $M$-ball. For instance,

$$\Gamma_{\text{IGD}} = \arg\max_\Gamma \sum_{u \in \mathcal{U}_S} \underbrace{\left\langle \Gamma \mathbf{g}^{(r)}, g_u^{(r)} \right\rangle}_{\text{Gradient matching}} - \gamma \underbrace{\left( \|\Gamma \mathbf{g}^{(r)} - g_{\text{FL}}^{(r)}\|^2 - \kappa \|g_{\text{FL}}^{(r)}\|^2 \right)}_{\text{Searching space limitation}}, \tag{11}$$

where $g_{\text{FL}}^{(r)}$ represents the gradients of the referenced FL methods (e.g., FedAvg (McMahan et al., 2017), where $g_{\text{FL}}^{(r)} = \sum_{u \in \mathcal{U}_S} N_u g_u^{(r)} / \sum_{u \in \mathcal{U}_S} N_u$). Beside the aforementioned first term, Eq. (11) consists of searching space limitation. Specifically, searching space limitation is activated by constraining the searching radius, preventing it from diverging too far from the specific range (i.e., $g_{\text{IGD}}^{(r)} \in \mathcal{B}_M(g_{\text{FL}}^{(r)}; \kappa \|g_{\text{FL}}^{(r)}\|^2)$, where $\mathcal{B}_M(g_{\text{FL}}^{(r)}; \kappa \|g_{\text{FL}}^{(r)}\|^2)$ is a $M$-ball centered at $g_{\text{FL}}^{(r)}$ and having

radius $\kappa \|g_{FL}^{(r)}\|^2$). Note that, the searching space limitation term reduces to $\arg\max_\Gamma \kappa \|g_{FL}^{(r)}\|\|g_\Gamma^{(r)}\|$ (after the relaxation in Theorem 1), which corresponds to the denominator in the application of cosine similarity. By constraining the search space to remain close to the reference FL gradient, the computational overhead of on-server optimization can be significantly reduced. This is because the optimization process requires fewer iterations to converge to optimal results. Additionally, this constraint mitigates the risk of the optimization process converging to suboptimal solutions.

**Pareto-based Optimization.** At every iteration, Eq. (11) consists of loop over $U_\mathcal{S}$ user, which requires $\mathcal{O}(U_\mathcal{S})$ computation complexity. To reduce the objective's complexity, we consider Pareto front, which provides a trade-off among the different objectives. We have the following definitions:

**Definition 2 (Pareto dominance (Zitzler & Thiele, 1999))** *Let $\theta^a, \theta^b \in R^M$ be two points, $\theta^a$ is said to dominate $\theta^b$ ($\theta^a \succ \theta^b$) if and only if $\mathcal{E}_u(\theta^a) \leq \mathcal{E}_u(\theta^b), \forall u \in \mathcal{U}$ and $\mathcal{E}_v(\theta^a) < \mathcal{E}_v(\theta^b), \exists v \in \mathcal{U}$.*

**Definition 3 (Pareto optimality (Zitzler & Thiele, 1999))** *$\theta^*$ is a Pareto optimal point and $\mathcal{E}(\theta^*)$ is a Pareto optimal objective if it does not exist $\hat{\theta} \in R^M$ satisfying $\hat{\theta} \succ \theta^*$. The set of all Pareto optimal points is called the Pareto set. The projection of the Pareto set onto the loss space is called the Pareto front.*

Leveraging Definitions 2 and 3, we have the following lemma:

**Lemma 2** *The average cosine similarity between given gradient vector $g_{IGD}^{(r)}$ and the domain-specific gradient is lower-bounded by the worst case cosine similarity as follows:*

$$\frac{1}{U} \sum_{u \in \mathcal{U}_\mathcal{S}} \left\langle g_u^{(r)}, g_{IGD}^{(r)} \right\rangle \geq \min_{u \in \mathcal{U}_\mathcal{S}} \left\langle g^{(r)}, g_{IGD}^{(r)} \right\rangle.$$

Lemma 2 allows us to realize that Eq. (11) can be reduced to maximizing the worst-case scenario. Thus, we have the surrogate FDG update as follows:

$$\theta_g^{(r)} = \theta_g^{(r-1)} - \eta_g g_{IGD}^{(r)} = \theta_g^{(r-1)} - \eta_g \Gamma_{IGD} \mathbf{g}^{(r)}, \tag{12a}$$

$$\text{s.t.} \quad \Gamma_{IGD} = \arg\max_\Gamma \min_{u \in \mathcal{U}_\mathcal{S}} \left[ \left\langle \Gamma\mathbf{g}^{(r)}, g_u^{(r)} \right\rangle - \gamma\left( \|\Gamma\mathbf{g}^{(r)} - g_{FL}^{(r)}\|^2 - \kappa\|g_{FL}^{(r)}\|^2 \right) \right]. \tag{12b}$$

However, solving Eq. (12) is complex due to the min-max problems with two variables. Thus, we simplify the optimization problem Eq. (12) as follows:

**Theorem 1 (FedOMG solution)** *Given $\Gamma = \{\gamma_u^{(r)} | u \in \mathcal{U}_\mathcal{S}, \sum_{u \in \mathcal{U}_\mathcal{S}} \gamma_u^{(r)} = 1\}$ is the set of learnable coefficients at each round $r$. Invariant gradient direction $g_{IGD}^{(r)}$ is characterized as follows:*

$$g_{IGD}^{(r)} = g_{FL}^{(r)} + \frac{\kappa\|g_{FL}^{(r)}\|}{\|\Gamma^*\mathbf{g}^{(r)}\|}\Gamma^*\mathbf{g}^{(r)} \quad s.t. \quad \Gamma^* = \arg\min_\Gamma \Gamma\mathbf{g}^{(r)} \cdot g_{FL}^{(r)} + \kappa\|g_{FL}^{(r)}\|\|g_\Gamma^{(r)}\|. \tag{13}$$

Proof of Theorem 1 is detailed in Appendix F.5. Theorem 1 provides an alternative solution for aggregating local gradients at the server. By relaxing Eq. (12) into Theorem 1, the optimization process becomes simplified. Instead of requiring iterative computations over all participating users in $\arg\max_\Gamma \min_{u \in \mathcal{U}_\mathcal{S}} \langle \Gamma\mathbf{g}^{(r)}, g_u^{(r)} \rangle$, the optimization can be performed through a straightforward computation, e.g., $\arg\min_\Gamma \Gamma\mathbf{g}^{(r)} \cdot g_{FL}^{(r)}$. The detailed algorithm is demonstrated as in Alg. 1

**Integratability.** $g_{FL}^{(r)}$ represents the gradients of referenced FL algorithm, while the local gradients $\mathbf{g}_u^{(r)}$ depend on that specific FL algorithm. Consequently, by using Theorem 1, we can integrate our on-server optimization into other FL algorithms. Furthermore, Theorem 1 also indicates that the IGD can be reduced to the chosen FL algorithms if we choose appropriate hyper-parameters:

**Corollary 1** *When the radius of the $\kappa$-hypersphere reduces to $0$, the IGD is reduced to the referenced FL algorithm. For instance, $\lim_{\kappa \to 0} g_{IGD} = g_{FL}^{(r)}$.*

## 5 THEORETICAL ANALYSIS

To prove the generalization capability of FedOMG, we have the following lemma:

**Lemma 3** *For any $\theta \in \Theta$, the domain divergence $d_{\mathcal{H}\triangle\mathcal{H}}(A, B)$ is bounded by the expectation of gradient divergence between domain $A$ and domain $B$.*

$$d_{\mathcal{H}\triangle\mathcal{H}}(A, B) \leq \frac{1}{\mu} d_{\mathcal{G}\circ\theta}(A, B), \tag{14}$$

*where $d_{\mathcal{G}\circ\theta}(A, B)$ is the gradient divergence of model $\theta$ when training in two domains $A$ and $B$.*

Proof of Lemma 3 is detailed in Appendix F.3. Lemma 3 establishes the connection between domain divergence and the gradient divergence it induces. Consequently, the domain shift $d_{\mathcal{H}\triangle\mathcal{H}}(\mathcal{D}_u, \mathcal{D}_v)$ from Eq. (1) can be interpreted as gradient divergence, which serves as the primary minimization objective in our work. We then prove that the generalization gap on an unseen domain $\mathcal{D}_{\mathcal{T}}$ by the optimal solution on $\mathcal{D}_{\mathcal{S}}$ is upper-bounded by the generalization gap in the source domains in Theorem 2. From Theorem 2, the domain generalization gap on an unseen domain $\mathcal{D}_{\mathcal{T}}$ is bounded by $X$ and the domain divergence $d_{\mathcal{H}\triangle\mathcal{H}}(\mathcal{D}_u, \mathcal{D}_v)$.

**Theorem 2** *Let $\theta^R$ denote the global model after $R$ rounds FL, $\theta_u^*$, and $\theta_{\mathcal{T}}^*$ mean the local optimal for each client and the unseen target domain, respectively. The local objectives follow the $\mu$ strongly convex from Assumption 2. For any $\delta \in (0, 1)$, the domain generalization gap for the unseen domain $\mathcal{D}_{\mathcal{T}}$ can be bounded by the following equation with a probability of at least $1 - \delta$.*

$$\mathcal{E}_{\mathcal{D}_{\mathcal{T}}}(\theta^R) - \mathcal{E}_{\mathcal{D}_{\mathcal{T}}}(\theta_{\mathcal{D}_{\mathcal{T}}}^*)$$

$$\leq \sum_{u \in \mathcal{U}_{\mathcal{S}}} \gamma_u \left[ \mathcal{E}_{\hat{\mathcal{D}}_u}(\theta) + \sum_{v \in \mathcal{U}_{\mathcal{S}}} \frac{d_{\mathcal{G}\circ\theta}(\hat{\mathcal{D}}_u, \hat{\mathcal{D}}_v)}{\mu} + d_{\mathcal{H}\triangle\mathcal{H}}(\mathcal{D}_{\mathcal{S}}, \mathcal{D}_{\mathcal{T}}) + \frac{\sqrt{\log \frac{M}{\delta}} + \sqrt{\log \frac{U_{\mathcal{S}}M}{\delta}}}{\sqrt{2N_u}} \right] + \zeta^*,$$

*where $\hat{\mathcal{D}}_u, \hat{\mathcal{D}}_v$ are the sampled counterparts from the domain $\mathcal{D}_u, \mathcal{D}_u$, respectively. $d_{\mathcal{G}\circ\theta}(\hat{\mathcal{D}}_u, \hat{\mathcal{D}}_v)$ denotes the gradient divergence of model $\theta$ when training on two different domains $\hat{\mathcal{D}}_u$ and $\hat{\mathcal{D}}_v$.*

Proof of Theorem 2 is detailed in Appendix F.4. The generalization gap at round $R$ on target domain $\mathcal{E}_{\mathcal{D}_{\mathcal{T}}}(\theta^R) - \mathcal{E}_{\mathcal{D}_{\mathcal{T}}}(\theta_{\mathcal{D}_{\mathcal{T}}}^*)$ affected by the divergence among domains' gradients $d_{\mathcal{G}\circ\theta}(\hat{\mathcal{D}}_u, \hat{\mathcal{D}}_v)$. These domain gradients are affected by the FedOMG algorithm, where our goal is to minimize the gradient divergence at every round. By combining Theorem 2 and Lemma 1 , we obtain the upper-boundary on the FedOMG.

**Corollary 2** *Assume FedOMG solution is given by $g_{IGD}^{(r)} = \arg\max_{\mathcal{G}} \sum_{u \in \mathcal{U}_{\mathcal{S}}} \sum_{v \in \mathcal{U}_{\mathcal{S}}} d_{\mathcal{G}\circ\theta}(\hat{\mathcal{D}}_u, \hat{\mathcal{D}}_v)$. Generalization gap of FedOMG is bounded by the followings with a probability of at least $1 - \delta$.*

$$\mathcal{E}_{\mathcal{D}_{\mathcal{T}}}(\theta^R) - \mathcal{E}_{\mathcal{D}_{\mathcal{T}}}(\theta_{\mathcal{D}_{\mathcal{T}}}^*) \leq \sum_{u \in \mathcal{U}_{\mathcal{S}}} \gamma_u \left[ \mathcal{E}_{\hat{\mathcal{D}}_u}(\theta) + \max_{\mathcal{G}} \frac{1}{\mu} \sum_{v \in \mathcal{U}_{\mathcal{S}}} d_{\mathcal{G}\circ\theta}(\hat{\mathcal{D}}_u, \hat{\mathcal{D}}_v) \right.$$

$$\left. + d_{\mathcal{H}\triangle\mathcal{H}}(\mathcal{D}_{\mathcal{S}}, \mathcal{D}_{\mathcal{T}}) + \frac{\sqrt{\log \frac{M}{\delta}} + \sqrt{\log \frac{U_{\mathcal{S}}M}{\delta}}}{\sqrt{2N_u}} \right] + \zeta^*,$$

Consequently, the gradient matching can reduce the FedOMG generalization gap every round, thus reducing the generalization gap on the unseen target data domain $\mathcal{D}_{\mathcal{T}}$.

## 6 EXPERIMENTAL EVALUATIONS

### 6.1 ILLUSTRATIVE TOY TASKS

Fig. 1 presents the performance of FedOMG on Rect-4, revealing several key findings. Firstly, a significant gradient conflict is observed when training in systems with heterogeneous users (e.g., feature divergence), while in the IID setting, the gradient is more aligned toward one direction.

Secondly, in FedAvg, users 1 and 2 tend to steer the global model away from the ideal ERM (where data is randomly sampled and not affected by heterogeneity). This issue is exacerbated in the FDG setting, causing the gradient to shift upward, meaning the bias increases (ideally, the bias should be close to zero). Thirdly, FedOMG demonstrates effective gradient matching, aligning closely with ERM gradients in the FL setting or moving towards the optimal point in the FDG setting. The details of experimental settings are demonstrated in Appendix C.

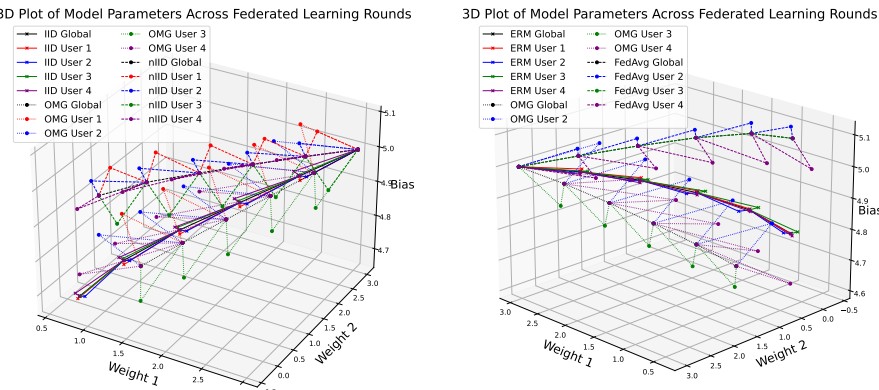

Figure 1: Illustrative toy task on two settings. 1) FL settings, where all users are participating in the training (left figure), 2) FDG setting: one user is excluded in the training (right figure).

## 6.2 EXPERIMENTAL MAIN RESULTS

**FL performance comparison with baselines.** Tab. 1 presents the results for our method in a conventional FL setting. The best result is highlighted in pink, the second-best result is in bold, and our proposed method is highlighted in yellow. We observe that our method consistently achieves the best or second-best performance across all datasets compared to other methods. Notably, our proposed algorithm demonstrates robustness, as it integrates effectively with other FL methods, resulting in superior performance. Full results are shown in Appendix E.1.

**Discussion on orthogonality of FedOMG with FL baselines.** Performances of the integration of FedOMG into three other FL algorithms (i.e., PerAvg, FedRod, FedBabu) are shown in Tab. 1. The integration of advanced methods consistently outperforms standard FL algorithms. Notably, the combination of FedRod and FedOMG yields a significant enhancement in performance. This improvement can be attributed to the complementary nature of these two techniques. FedRod introduces layer disentanglement, which enhances the personalization capabilities of FL clients. In contrast, FedOMG focuses on on-server gradient matching, which improves the overall generalization of the FL model. Together, these orthogonal approaches create a synergistic effect that boosts both personalization and generalization in FL.

Table 1: Comparison of methods across different datasets and non-IID scenarios. The setups are 100 users at 20% user participation (i.e. $\alpha \in \{0.1, 0.5\}$). Results are averaged after 5 times.

| Setting | Non-IID ($\alpha = 0.1$) | | | | Non-IID ($\alpha = 0.5$) | | | |
|---|---|---|---|---|---|---|---|---|
| Dataset
Method | MNIST | CIFAR10 | CIFAR100 | EMNIST | MNIST | CIFAR10 | CIFAR100 | EMNIST |
| FedAvg | $93.77 \pm 0.33$ | $60.89 \pm 0.12$ | $28.78 \pm 0.10$ | $86.45 \pm 0.30$ | $91.47 \pm 0.26$ | $59.12 \pm 0.21$ | $25.55 \pm 0.22$ | $84.23 \pm 0.18$ |
| PerAvg | $94.44 \pm 0.26$ | $64.76 \pm 0.13$ | $36.27 \pm 0.32$ | $87.87 \pm 0.17$ | $93.52 \pm 0.32$ | $64.46 \pm 0.02$ | $27.29 \pm 0.20$ | $84.87 \pm 0.01$ |
| FedRod | $98.09 \pm 0.27$ | $88.90 \pm 0.19$ | $44.33 \pm 0.26$ | $97.50 \pm 0.24$ | $96.85 \pm 0.11$ | $70.52 \pm 0.31$ | $28.17 \pm 0.12$ | $96.46 \pm 0.22$ |
| FedPac | $96.90 \pm 0.03$ | $87.81 \pm 0.17$ | $48.83 \pm 0.04$ | $97.74 \pm 0.32$ | $94.63 \pm 0.22$ | $73.02 \pm 0.16$ | $29.94 \pm 0.24$ | $94.10 \pm 0.09$ |
| FedBabu | $96.40 \pm 0.08$ | $88.19 \pm 0.19$ | $49.18 \pm 0.09$ | $96.76 \pm 0.31$ | $95.10 \pm 0.24$ | $70.91 \pm 0.26$ | $28.33 \pm 0.23$ | $93.11 \pm 0.02$ |
| FedAS | $97.91 \pm 0.22$ | $89.15 \pm 0.06$ | $50.37 \pm 0.18$ | $97.71 \pm 0.15$ | $96.78 \pm 0.33$ | $\mathbf{75.75 \pm 0.21}$ | $\mathbf{32.57 \pm 0.31}$ | $94.53 \pm 0.12$ |
| FedOMG | $\mathbf{98.75 \pm 0.02}$ | $90.40 \pm 0.03$ | $48.76 \pm 0.15$ | $\mathbf{98.91 \pm 0.02}$ | $\mathbf{98.58 \pm 0.09}$ | $72.68 \pm 0.02$ | $30.64 \pm 0.32$ | $\mathbf{98.41 \pm 0.02}$ |
| PerAvg+OMG | $95.04 \pm 0.26$ | $84.46 \pm 0.13$ | $44.52 \pm 0.32$ | $95.18 \pm 0.32$ | $94.52 \pm 0.32$ | $70.76 \pm 0.02$ | $28.29 \pm 0.20$ | $90.68 \pm 0.01$ |
| FedRod+OMG | $99.63 \pm 0.08$ | $95.81 \pm 0.04$ | $55.39 \pm 0.16$ | $99.62 \pm 0.03$ | $99.55 \pm 0.08$ | $76.80 \pm 0.12$ | $36.78 \pm 0.04$ | $99.10 \pm 0.63$ |
| FedBabu+OMG | $98.12 \pm 0.04$ | $\mathbf{92.19 \pm 0.09}$ | $\mathbf{51.27 \pm 0.09}$ | $98.62 \pm 0.20$ | $97.28 \pm 0.09$ | $75.41 \pm 0.13$ | $31.82 \pm 0.16$ | $97.25 \pm 0.02$ |

**FDG performance.** In Tab. 2 we report the results for domain generalization performance. Our FedOMG consistently outperforms other methods, i.e., $5\% - 6\%$ gain in target domain accuracy.

**Discussion on orthogonality of FedOMG with FDG baselines.** Performances of the integration of FedOMG into three other FDG algorithms (i.e., PerAvg, FedRod, FedBabu) are shown in Tab. 1. The integration of advanced methods consistently outperforms standard FDG algorithms. However, only FedSAM+OMG outperforms the standalone FedOMG model. This is because FedSAM and FedOMG address two orthogonal and fundamental aspects of FDG (i.e., FedSAM reduces the sharpness of the loss landscape, while FedOMG focuses on mitigating gradient divergences across domains). In contrast, the regularization techniques used in FedIIR and FedSR primarily aim to minimize domain divergence on the client side, a contribution that overlaps to some extent with FedOMG's approach.

Table 2: Domain generalization performance on VLCS, PACS, and OfficeHome Datasets

| Algorithm | VLCS | | | | | PACS | | | | | OfficeHome | | | | |
|---|---|---|---|---|---|---|---|---|---|---|---|---|---|---|---|
| | V | L | C | S | Avg | P | A | C | S | Avg | A | C | P | R | Avg |
| FedAvg | $72.5 \pm 0.8$ | $61.1 \pm 0.9$ | $93.6 \pm 1.0$ | $65.4 \pm 0.3$ | 73.1 | $92.7 \pm 0.6$ | $77.2 \pm 1.0$ | $77.9 \pm 0.5$ | $81.0 \pm 0.8$ | 82.7 | $57.7 \pm 0.9$ | $48.3 \pm 0.1$ | $72.8 \pm 0.2$ | $75.3 \pm 0.1$ | 63.5 |
| FedGA | $74.4 \pm 0.1$ | $56.9 \pm 1.0$ | $94.3 \pm 0.6$ | $68.9 \pm 0.9$ | 73.4 | $93.9 \pm 0.2$ | $81.2 \pm 0.7$ | $76.7 \pm 0.4$ | $82.5 \pm 0.1$ | 83.5 | $58.5 \pm 0.4$ | $54.3 \pm 0.6$ | $73.3 \pm 0.8$ | $74.7 \pm 1.0$ | 65.2 |
| FedSAM | $74.5 \pm 0.3$ | $58.0 \pm 0.4$ | $92.9 \pm 0.8$ | $74.1 \pm 0.7$ | 74.8 | $91.2 \pm 0.1$ | $74.4 \pm 0.9$ | $77.7 \pm 0.3$ | $83.3 \pm 0.2$ | 81.6 | $55.3 \pm 0.2$ | $54.7 \pm 0.4$ | $73.5 \pm 0.5$ | $73.7 \pm 0.7$ | 64.3 |
| FedIIR | $76.1 \pm 1.4$ | $60.9 \pm 0.2$ | $96.3 \pm 0.4$ | $73.2 \pm 0.8$ | 76.6 | $94.2 \pm 0.2$ | $82.9 \pm 0.8$ | $75.8 \pm 0.3$ | $81.9 \pm 0.8$ | 83.7 | $57.1 \pm 0.4$ | $49.8 \pm 0.6$ | $74.2 \pm 0.1$ | $76.1 \pm 0.1$ | 64.4 |
| FedSR | $72.8 \pm 0.3$ | $62.3 \pm 0.3$ | $93.8 \pm 0.5$ | $74.4 \pm 0.6$ | 75.8 | $94.0 \pm 0.6$ | $82.8 \pm 1.5$ | $75.2 \pm 0.5$ | $81.7 \pm 0.8$ | 83.4 | $57.9 \pm 0.2$ | $50.3 \pm 0.6$ | $73.3 \pm 0.1$ | $75.5 \pm 0.1$ | 64.3 |
| StableFDG | $73.6 \pm 0.1$ | $59.2 \pm 0.7$ | $98.1 \pm 0.2$ | $70.2 \pm 1.1$ | 75.3 | $94.8 \pm 0.1$ | $83.0 \pm 1.1$ | $79.3 \pm 0.2$ | $79.7 \pm 0.8$ | 84.2 | $57.1 \pm 0.3$ | $57.9 \pm 0.5$ | $72.7 \pm 0.6$ | $72.1 \pm 0.8$ | 65.0 |
| FedOMG | $82.3 \pm 0.5$ | $67.5 \pm 0.4$ | $99.3 \pm 0.1$ | $79.1 \pm 0.5$ | 82.0 | $98.0 \pm 0.2$ | $89.7 \pm 0.4$ | $81.4 \pm 0.8$ | $84.3 \pm 0.5$ | 88.4 | $65.4 \pm 0.4$ | $58.1 \pm 0.3$ | $77.5 \pm 0.4$ | $78.9 \pm 0.5$ | 70.0 |
| FedIIR+OMG | $75.3 \pm 1.3$ | $64.0 \pm 0.2$ | $97.7 \pm 0.1$ | $72.8 \pm 0.2$ | 77.5 | $97.7 \pm 0.1$ | $83.0 \pm 1.1$ | $80.8 \pm 0.2$ | $79.3 \pm 0.3$ | 85.2 | $62.0 \pm 0.3$ | $52.8 \pm 0.5$ | $74.3 \pm 0.6$ | $76.9 \pm 0.8$ | 66.5 |
| FedSAM+OMG | $82.7 \pm 0.7$ | $69.4 \pm 0.9$ | $99.3 \pm 0.3$ | $78.5 \pm 0.8$ | 82.5 | $98.3 \pm 0.1$ | $88.9 \pm 1.2$ | $82.7 \pm 0.3$ | $85.5 \pm 0.2$ | 88.8 | $65.8 \pm 0.2$ | $58.9 \pm 0.4$ | $78.9 \pm 0.5$ | $79.3 \pm 0.7$ | 70.9 |
| FedSR+OMG | $73.6 \pm 0.1$ | $66.0 \pm 0.3$ | $94.8 \pm 0.2$ | $73.3 \pm 3.3$ | 76.9 | $97.2 \pm 0.1$ | $83.2 \pm 1.1$ | $79.8 \pm 0.2$ | $79.3 \pm 3.3$ | 84.8 | $61.7 \pm 0.3$ | $53.3 \pm 0.5$ | $73.6 \pm 0.6$ | $75.9 \pm 0.8$ | 66.1 |

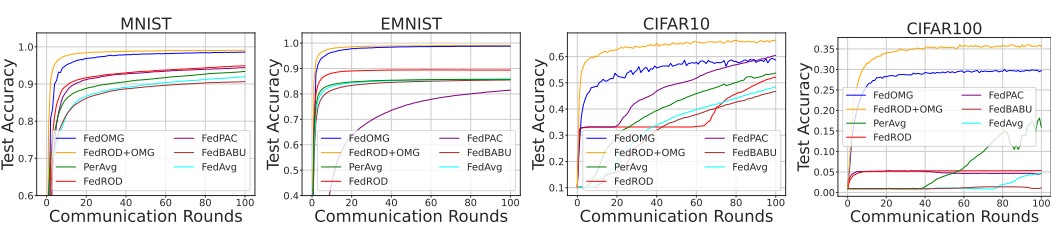

Figure 2: Performance comparison without pretrained models for $\alpha = 1, U = 20$.

## 6.3 ABLATION STUDIES

**Effects of training FL without pretrained models:** Fig. 2 demonstrates the FL training without pretrained models. FedOMG results in the more stable convergence than most other FL baselines along with FedPAC and FedRod. Notably, the combination of FedOMG and FedRod shows a substantial improvement in performance compared to other baselines. In summary, FedOMG, both individually and in combination with other FL methods, offers enhanced robustness when training FL models from scratch without relying on pretrained models.

**Global learning rate $\eta$:** In Fig. 3, we present FedOMG results with varied $\eta$, and fixed $\kappa$ to 0.5, and each result is the average of three runs. In EMNIST and CIFAR-10, the gradient matching task is straightforward. Therefore, higher $\eta$ accelerates convergence to the optimal state. However, on more challenging datasets with domain shift issues, i.e., PACS, VLCS, and OfficeHome, a lower learning rate proves to be more effective.

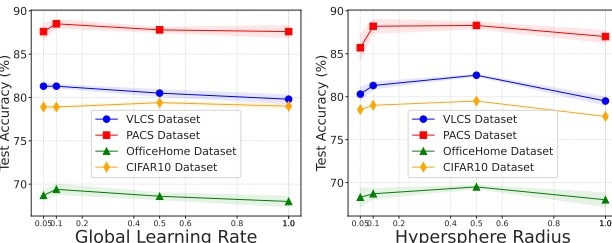

Figure 3: Global LR $\eta$.  Figure 4: Searching ratio $\kappa$.

**Searching radius ratio $\kappa$:** In this evaluation, we test our method with four different values of $\gamma$ across seven benchmark datasets. Each experiment is run three times, and the results are averaged. As shown in Fig. 4, FedOMG performs optimally when the radius is set to $\kappa \approx 0.5$. When $\kappa \to 0$, the performance leans towards the FedAvg, while setting a high $\kappa$, the performance degrades gradually as the searching space become large, thus, it is struggled to find the optimal solution.

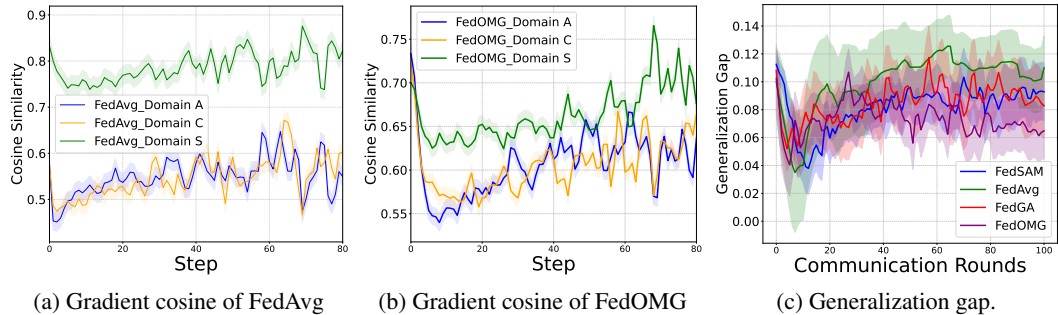

(a) Gradient cosine of FedAvg     (b) Gradient cosine of FedOMG     (c) Generalization gap.

Figure 5: The evaluations on gradient invariance. Algorithms with stronger invariance properties result in smaller gaps in cosine similarity between each domain's and the global gradients. The algorithms are evaluated on PACS dataset, where source domains are A, C, S, and target domain is P.

**Evaluation on gradient invariance:** To assess the invariant property of FedOMG, we calculate the cosine similarity between the gradients of each source domain and the global gradient, and visualize in Figs. 5a, 5b. As it can be seen from the figures, the significance of FedOMG compared to FedAvg is two-fold. First of all, the cosine similarities of all domains in FedOMG hold a low variance among domains. This mean that all of the domains share the same divergence compared with the global gradients. Secondly, the overwhelmingly high cosine value in domain S of FedAvg may results from the overfitting to domain S. As a consequence, the FedAvg significantly lacks generalization capability compared to that of the FedOMG (see Fig. 5c).

**Discussion on impact of local epochs and local learning rate:** Fig. 6 illustrates the fine-tuning of FedOMG under varying numbers of local training epochs and learning rates. The results indicate that FedOMG reaches optimal performance when the number of local epochs is set to 5. This suggests that after 5 epochs, the local gradients adequately represent the users' gradient direction during each round. Extending the local training duration beyond this point incurs higher computational overhead while reducing the generalization of on-server gradient matching, potentially due to overfitting to individual user datasets. Furthermore, the optimal performance is observed when the local learning rate is set to 0.001.

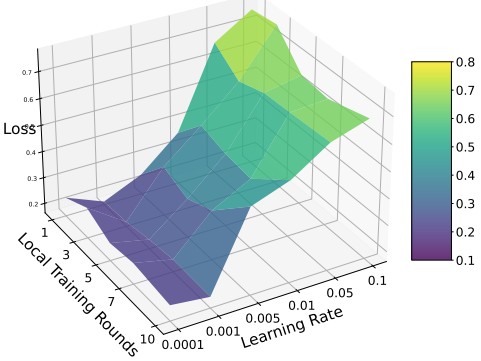

Figure 6: Evaluations on local training epochs and learning rate.

**Additional results.** Detailed results when training with different target domain on VLCS, PACS, OfficeHome are demonstrated in Appendix E.3. Results on computation cost are demonstrated in Appendix E.4. Additional results conducted on FDG benchmark (Bai et al., 2024) are demonstrated in Appendix E.5.

## 7 CONCLUSION

In this paper, we present FedOMG, a novel and effective federated domain generalization method aimed at addressing the challenge of heterogeneous client data distributions by learning an invariant gradient. Our method introduces two key innovations: (1) utilizing the local gradient as a critical factor for server-side optimization, and (2) devising a training strategy that identifies optimal coefficients to approximate the invariant gradient across client source domains. We provide theoretical proof to ensure the global invariant gradient solution and conduct extensive experiments demonstrating FedOMG's significant improvements in both standard FL and domain generalization (FDG) settings. This work establishes an optimal approach for server optimization, leading to enhanced FL performance while maintaining privacy constraints.

ACKNOWLEDGMENTS

This work was supported by Institute of Information & communications Technology Planning & Evaluation (IITP) under the Artificial Intelligence Convergence Innovation Human Resources Development (IITP-2025-RS-2023-00254177) grant funded by the Korea government(MSIT). This research was supported by the MSIT(Ministry of Science and ICT), Korea, under the ITRC(Information Technology Research Center) support program(IITP-2025-RS-2023-00260098) supervised by the IITP(Institute for Information & Communications Technology Planning & Evaluation). This work was supported by the National Research Foundation of Korea(NRF) grant funded by the Korea government(MSIT)(RS-2024-00336962). The work of Quoc-Viet Pham is supported in part by the European Union under the ENSURE-6G project (Grant Agreement No. 101182933).

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

**Appendix**

The appendix is organised as follows:

- Appendix A provides the literature reviews.

- Appendix B provides details of proposed algorithm.

- Appendix C provides toy dataset description.

- Appendix D provides extensive experiment settings and hyper-parameters.

- Appendix E provides additional results for standard FL and ablation test for generalization performance.

- Appendix F provides proof on theorems.

## A  RELATED WORKS

**Federated Learning:** To contextualize our paper's contribution to on-server optimization, FL can be divided into two primary categories: 1) local training and 2) on-server aggregation. In the first category, substantial efforts have been dedicated to improving the performance of FL, particularly in scenarios involving non-IID data. FedRod (Chen & Chao, 2022) proposes using balanced softmax for training a generic model while utilizing vanilla softmax for personalized heads. FedBABU (Oh et al., 2022) keeps the global classifier fixed during feature representation learning and performing local adaptation through fine-tuning. FedPAC (Xu et al., 2023) adds a regularizer on local training to learn task-invariant representations. Similar to FedPAC, FedAS (Yang et al., 2024) finds the shared parameters which are task-invariant by leveraging the parameter alignment technique. Recently, researchers have focused on on-server aggregation to improve the performance of FL. This approach can be implemented by utilizing generative AI on the server with pseudo data (Zhu et al., 2021), or leveraging local gradients to adjust the learning rate (Jhunjhunwala et al., 2023; Panchal et al., 2023).

**Federated Domain Generalization and Adaptation:** Huang et al. (2022) and Huang et al. (2024) leverage unlabeled public data to reduce bias toward local data distributions. Nguyen et al. (2022a) regularizes local training via conditional mutual information. Guo et al. (2023) adds a regularization on the local training to implicitly align gradients between the local global models. Qu et al. (2022), Sun et al. (2023) and Fan et al. (2024) leverage sharpness aware minimization to smooth the loss landscape. Tang et al. (2024) utilizes the shared local hidden features to reduce the gradient dissimilarity. Zhang et al. (2023b) introduces server-side weight aggregation adjustments leveraging the auxiliary local generalization gap. Jiang et al. (2024) utilizes the divergence between source and target data gradients to formulate a joint server aggregation rule. However, it requires access to target domain data, which may not be feasible in domain generalization scenarios. Park et al. (2023) introduced a style-based strategy to enhance the diversity of data on source client. Additionally, they integrated attention into the network to highlight common and important features across different domains.

## B  DETAILED ALGORITHMS

In Algorithm 1, we present the detailed training procedure of our FedOMG approach. The primary contribution of our method lies in the server-side optimization, while the client-side follows the standard FedAvg algorithm (McMahan et al., 2017), ensuring no additional communication or computation overhead. During each communication round, clients receive the global model parameters and perform local training using SGD for $E$ epochs. The locally updated parameters are then sent back to the centralized server for optimization. On the server, *local gradients* are computed and used as key components in solving Eq. (15). The resulting gradient closely approximates the invariant gradient direction, as demonstrated in Section 4. Subsequently, the global gradient is computed as shown in Eq. (16), and the global model is updated according to Eq. (17).

---

**Algorithm 1:** Federated Learning via On-server Matching Gradient

---

**Input:** set of source clients $\mathcal{U}_\mathcal{S}$, number of communication rounds $R$, local learning rate $\eta$, global learning rate $\eta_g$, searching space hyper-parameter $\kappa$.

**Output:** $\theta_g^{(R)}$

**Clients Update:**

**for** *client $u \in \mathcal{U}_\mathcal{S}$* **do**

    **Receive** global model $\theta_u^{(r)} = \theta_g^{(r)}$;

    **for** *local epoch $e \in E$* **do**

        Sample mini-batch $\zeta$ from local data $\mathcal{D}_u$;

        Calculate gradient $\nabla\mathcal{E}(\theta_u^{(r,e)}, \zeta)$;

        Update client's model: $\theta_u^{(r,e+1)} = \theta_u^{(r,e)} - \eta\nabla\mathcal{E}(\theta_u^{(r,e)}, \zeta)$;

    **end for**

    Upload client's model $\theta_u^{(r,E)}$ to server;

**end for**

**Server Optimization:**

**for** *round $r = 0, \ldots, R$* **do**

    **Clients Updates**;

    Calculate $g_u^{(r)} = \theta_u^{(r,E)} - \theta_u^{(r)}$, $\mathbf{g}^{(r)} = \{g_u^{(r)} | u \in \mathcal{U}_\mathcal{S}\}$;

    Calculate $g_{FL}^{(r)}$ (e.g., $g_{FL}^{(r)} = \frac{1}{U}\sum_{u=1}^{U} g_u^{(r)}$ as the FedAvg update);

    Solve for $\Gamma^*$:

$$\Gamma^* = \arg\min_{\Gamma} \Gamma\mathbf{g}^{(r)} \cdot g_{\text{FL}}^{(r)} + \kappa\|g_{\text{FL}}^{(r)}\|\|\Gamma\mathbf{g}^{(r)}\|, \tag{15}$$

    Update the model:

$$g_{\text{IGD}}^{(r)} = g_{\text{FL}}^{(r)} + \frac{\kappa\|g_{\text{FL}}^{(r)}\|}{\|\Gamma^*\mathbf{g}^{(r)}\|}\Gamma^*\mathbf{g}^{(r)}, \tag{16}$$

$$\theta_g^{(r)} = \theta_g^{(r-1)} - \eta_g g_{\text{IGD}}^{(r)}. \tag{17}$$

**end for**

---

## C TOY DATASET DESCRIPTION

**Dataset and Settings Descriptions.** Based on Zhao et al. (2019), we design a toy dataset, coined Rect-4, a synthetic binary classification dataset with 4 domains, according to 4 different users. In each domain, a two-dimensional key point $x_d = (x_{d,1}, x_{d,2})$ is randomly selected in the two-dimensional space with varying region distributions. To visualize the gradient of the toy dataset, we design a 1-layer, 2-parameters network. To this end, we can visualize the gradients in a 3-D space, consisting of 2 parameters and 1 weight.

We visualize the training data in Figs. 7 and 8. In the FDG setting, the users are from different domains. To this end, we design the data where the point are distributed into rectangular with different size and shape. The rationale of designing the data distribution is as follows:

- The global dataset consists of two classes from two rectangular regions, which has the classification boundary equal to $y = 0$.

- Each domain-wise dataset has different classification boundary (e.g., $x = -6$ for domain 1). We add the noisy data on every domains so that the user assign to each domain will tend to learn the local boundary instead of the global boundary. Thus, we can observe the gradient divergence more clearly, as the global boundary is not the optimal solution when learn on local dataset.

- All of the local classification boundary is orthogonal from the global classification boundary, thus, we can make the learning more challenging despite the simplicity of the toy dataset.

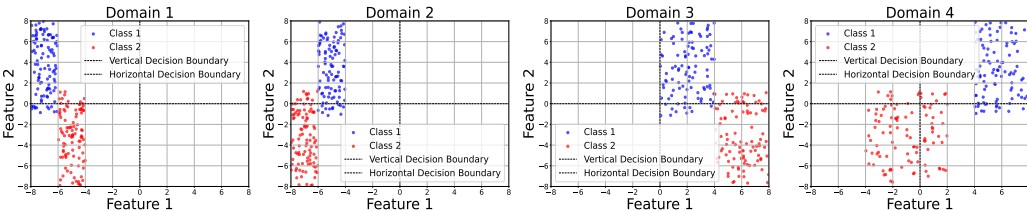

Figure 7: Illustration of users with different domains.

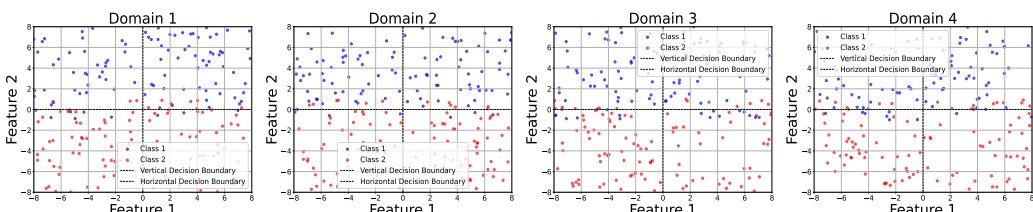

Figure 8: Illustration of users with same domains.

# D    EXTENSIVE EXPERIMENTAL SETTINGS

## D.1    DATASETS

**MNIST** (Lecun et al., 1998) consists of 70,000 grayscale images of handwritten digits and ten classes

**EMNIST** (Cohen et al., 2017) extension version of original MNIST dataset consists of 70,000 samples and 10 classes

**CIFAR10** (Krizhevsky, 2012) consists of 60,000 images across 10 classes "Airplane", "Automobile", "Bird", "Cat", "Deer", "Dog", "Frog", "Horse", "Ship" and "Truck"

**CIFAR-100** (Krizhevsky, 2012) is a more challenging extension of CIFAR-10, consisting of 60,000 images distributed across 100 distinct classes.

**VLCS** (Torralba & Efros, 2011) includes 10,729 images from four domains: "VOC2007", "LabelMe", "Caltech101", "SUN09" and five classes ('bird', 'car', 'chair', 'dog' and 'person').

**PACS** (Li et al., 2017) includes 9,991 images from four domains: "Photos", "Art", "Cartoons", and "Sketches" and seven classes ('dog', 'elephant', 'giraffe', 'guitar', 'horse', 'house', and 'person').

**OfficeHome** (Venkateswara et al., 2017) includes four domains: "Art", "Clipart", "Product", and "Real". The dataset contains 15,588 samples and sixty five classes.

## D.2    BASELINES

**FL Baselines.** We consider the following baselines: FedAvg (McMahan et al., 2017), PerAvg (Fallah et al., 2020), FedRod (Chen & Chao, 2022), FedBABU (Oh et al., 2022), FedPac (Xu et al., 2023) and FedAS (Yang et al., 2024) comparing with our proposed method FedOMG.

**FDG Baselines.** We implement the following methods: FedAvg (McMahan et al., 2017), FedGA (Zhang et al., 2023b), FedIIR (Guo et al., 2023), FedSam (Qu et al., 2022), FedSR (Nguyen et al., 2022a) and StableFDG (Park et al., 2023).

## D.3    EVALUATION METRIC

We report accuracy as the standard metric, defined as the ratio of correctly paired samples to the total number of samples, with Top-1 accuracy being used. In FL scenarios, accuracy is reported as the ratio of the total number of correct samples across all clients to the total number of samples. In FDG scenarios, accuracy is reported for each test domain individually, and the average accuracy across all domains is provided.

## D.4    MODEL

For the standard FL task, we use a CNN architecture consisting of two convolutional layers (32 and 64 channels) followed by max pooling layers. It includes two fully connected layers with 512 and number of output classes units, respectively, and ends with a softmax output for class probabilities.

For the domain generalization task, we employ a pretrained ResNet-18 backbone model (Zhang et al., 2023b; Park et al., 2023; Guo et al., 2023), which is downloaded from PyTorch. The detailed architecture of the ResNet-18 model used is outlined in Tab. 3.

Table 3: Summary of model architecture of ResNet-18 model for FDG.

| Layer Name | Number of Layers | Parameters (M) |
|---|---|---|
| Conv1 | 1 | 0.009 |
| Conv2_x (Residual Blocks) | 4 | 0.073 |
| Conv3_x (Residual Blocks) | 4 | 0.231 |
| Conv4_x (Residual Blocks) | 4 | 0.919 |
| Conv5_x (Residual Blocks) | 4 | 3.673 |
| Fully Connected | 1 | 0.513 |
| **Total** | **18** | **11.69M** |

## D.5 Hyper-parameters

In this section, we present the hyper-parameters chosen for FedOMG across different datasets.

Table 4: Hyper-parameter Summary on PACS, VLCS, and OfficeHome

| Method | Hyper-parameter | PACS | VLCS | OfficeHome |
|---|---|---|---|---|
| FedOMG | Global lr | $5 \times 10^{-2}$ | $5 \times 10^{-2}$ | $5 \times 10^{-2}$ |
| | Training lr | 25 | 25 | 25 |
| | Iteration | 21 | 21 | 21 |
| | Momentum | 0.5 | 0.5 | 0.5 |
| | Searching radius | 0.5 | 0.5 | 0.5 |
| FedSR+OMG | Global lr | $5 \times 10^{-2}$ | $5 \times 10^{-2}$ | $5 \times 10^{-2}$ |
| | Training lr | 25 | 25 | 25 |
| | Iteration | 21 | 21 | 21 |
| | Momentum | 0.5 | 0.5 | 0.5 |
| | Searching radius | 0.5 | 0.5 | 0.5 |
| | L2 Regularizer | 0.01 | 0.01 | 0.01 |
| | Cmi Regularizer | 0.001 | 0.001 | 0.001 |
| FedIIR+OMG | Hyper-parameter $\gamma$ | $1 \times 10^{-4}$ | $1 \times 10^{-4}$ | $1 \times 10^{-4}$ |
| | Global lr | $5 \times 10^{-2}$ | $5 \times 10^{-2}$ | $5 \times 10^{-2}$ |
| | Training lr | 25 | 25 | 25 |
| | Iteration | 21 | 21 | 21 |
| | Momentum | 0.5 | 0.5 | 0.5 |
| | Searching radius | 0.5 | 0.5 | 0.5 |
| FedSAM+OMG | Global lr | $5 \times 10^{-2}$ | $5 \times 10^{-2}$ | $5 \times 10^{-2}$ |
| | Training lr | 25 | 25 | 25 |
| | Iteration | 21 | 21 | 21 |
| | Momentum | 0.5 | 0.5 | 0.5 |
| | Searching radius | 0.5 | 0.5 | 0.5 |
| | Perturbation control $\rho$ | 0.6 | 0.6 | 0.6 |
| | momentum parameter $\beta$ | 0.6 | 0.6 | 0.6 |

## D.6 Implementation Details

To evaluate the performance of our proposed FDG method, we adopt the methodologies described in Lin et al. (2020); Acar et al. (2021) to simulate non-IID data. We utilize the Dirichlet distribution with two levels of data heterogeneity, specifically $\alpha = 0.1$ and $\alpha = 0.5$. Our experimental setup comprises three configurations of 100 clients, with join rate ratios of 1, 0.6, and 0.4, respectively, and global communication rounds set at 800. Each client undergoes 5 local training rounds using a local learning rate of 0.005, an SGD optimizer, and a batch size of 16. To ensure a fair comparison, all methods are evaluated with the same network architecture and settings.

For FDG scenarios, we evaluate model performance using the conventional *leave-one-domain-out* method (Guo et al., 2023; Fan et al., 2024), where one domain is designated as the test domain and all other domains are used for training. Number of clients is set to match the number of source domains. In line with Zhang et al. (2023b); Park et al. (2023); Guo et al. (2023), we use a pretrained ResNet-18 backbone model. Our experiments involve 100 global communication rounds with 5 local training rounds, utilizing the SGD optimizer with a learning rate of 0.001 and a training batch size of 16. Each experiment is conducted three times, and we report the average performance of the global model on the test domain, with each domain serving as the test domain once.

## E Additional Results and Ablation Tests

In this section, we provide additional results and conduct an ablation test, including (1) FL accuracy performance across different datasets, (2) convergence analysis, (3) the effect of using a pretrained model, and (4) an ablation test on various global learning rates and hyper-sphere radius.

Table 5: Comparison of methods across different datasets and non-IID scenarios.

| Problem | Non-IID ($\alpha = 0.1$) | | | | Non-IID ($\alpha = 0.5$) | | | |
|---|---|---|---|---|---|---|---|---|
| Dataset
Method | MNIST | CIFAR10 | CIFAR100 | EMNIST | MNIST | CIFAR10 | CIFAR100 | EMNIST |
| Number of clients $U = 100$ and participation ratio = 0.4 | | | | | | | | |
| FedOMG (Ours) | **98.07 ± 0.14** | 80.47 ± 0.03 | 42.31 ± 0.40 | **98.41 ± 0.02** | **96.84 ± 0.06** | 68.94 ± 0.17 | 29.64 ± 0.11 | **97.14 ± 0.03** |
| FedOMG+Rod | 99.34 ± 0.04 | 86.74 ± 0.05 | 51.70 ± 0.23 | 99.31 ± 0.01 | 98.90 ± 0.11 | 75.08 ± 0.12 | 33.97 ± 0.14 | 99.06 ± 0.01 |
| PerAvg | 89.18 ± 0.28 | 61.93 ± 0.23 | 30.52 ± 0.12 | 87.18 ± 0.24 | 87.04 ± 0.11 | 53.00 ± 0.12 | 24.09 ± 0.04 | 83.53 ± 0.12 |
| FedRod | 97.73 ± 0.17 | 82.09 ± 0.12 | 40.18 ± 0.10 | 95.25 ± 0.20 | 94.88 ± 0.29 | 68.03 ± 0.32 | 27.33 ± 0.16 | 92.46 ± 0.20 |
| FedPac | 95.05 ± 0.28 | 81.74 ± 0.15 | 40.78 ± 0.18 | 92.92 ± 0.09 | 93.36 ± 0.20 | 70.87 ± 0.25 | 28.56 ± 0.26 | 91.44 ± 0.05 |
| FedBabu | 94.92 ± 0.17 | 81.55 ± 0.18 | 41.30 ± 0.16 | 93.18 ± 0.13 | 93.62 ± 0.28 | 67.83 ± 0.02 | 28.06 ± 0.19 | 90.48 ± 0.18 |
| FedAS | 95.98 ± 0.24 | **83.35 ± 0.06** | **46.20 ± 0.76** | 93.71 ± 0.15 | 94.18 ± 0.33 | **72.29 ± 0.21** | 31.21 ± 0.31 | 92.53 ± 0.12 |
| FedAvg | 88.04 ± 0.22 | 55.75 ± 0.01 | 25.28 ± 0.05 | 85.83 ± 0.07 | 85.76 ± 0.17 | 49.44 ± 0.27 | 22.73 ± 0.19 | 82.57 ± 0.26 |
| Number of clients $U = 100$ and participation ratio = 0.6 | | | | | | | | |
| FedOMG (Ours) | **98.57 ± 0.15** | 86.50 ± 0.02 | 44.44 ± 0.10 | **98.79 ± 0.10** | **97.90 ± 0.05** | 70.33 ± 0.19 | 29.95 ± 0.21 | **97.92 ± 0.07** |
| FedOMG+Rod | 99.55 ± 0.15 | 93.72 ± 0.23 | 54.69 ± 0.12 | 99.61 ± 0.03 | 98.93 ± 0.11 | 75.84 ± 0.12 | 34.51 ± 0.10 | 99.08 ± 0.01 |
| PerAvg | 92.36 ± 0.24 | 62.19 ± 0.23 | 30.93 ± 0.01 | 87.34 ± 0.09 | 89.44 ± 0.24 | 60.74 ± 0.32 | 25.53 ± 0.26 | 83.74 ± 0.22 |
| FedRod | 97.92 ± 0.27 | 86.42 ± 0.09 | 40.39 ± 0.20 | 95.51 ± 0.09 | 95.82 ± 0.32 | 68.14 ± 0.32 | 28.56 ± 0.09 | 92.92 ± 0.28 |
| FedPac | 96.64 ± 0.30 | 86.35 ± 0.07 | 42.08 ± 0.21 | 93.92 ± 0.07 | 94.24 ± 0.30 | 72.23 ± 0.19 | 28.98 ± 0.04 | 91.55 ± 0.10 |
| FedBabu | 95.46 ± 0.25 | 86.36 ± 0.03 | 45.15 ± 0.29 | 93.58 ± 0.18 | 94.03 ± 0.26 | 69.52 ± 0.02 | 28.18 ± 0.12 | 90.03 ± 0.12 |
| FedAS | 97.12 ± 0.12 | **87.35 ± 0.13** | **48.46 ± 0.21** | 93.91 ± 0.15 | 95.18 ± 0.13 | **73.75 ± 0.31** | **31.70 ± 0.11** | 93.53 ± 0.12 |
| FedAvg | 90.17 ± 0.22 | 56.35 ± 0.06 | 27.46 ± 0.14 | 85.94 ± 0.15 | 87.90 ± 0.33 | 53.42 ± 0.21 | 24.07 ± 0.31 | 83.71 ± 0.19 |
| Number of clients $U = 100$ and participation ratio = 1 | | | | | | | | |
| FedOMG (Ours) | **98.75 ± 0.02** | 90.40 ± 0.03 | 48.76 ± 0.15 | **98.91 ± 0.02** | **98.58 ± 0.09** | 72.68 ± 0.02 | 30.64 ± 0.32 | **98.41 ± 0.02** |
| FedOMG+Rod | 99.63 ± 0.08 | 95.81 ± 0.04 | 55.39 ± 0.16 | 99.62 ± 0.03 | 99.55 ± 0.08 | 76.80 ± 0.12 | 36.78 ± 0.04 | 99.10 ± 0.63 |
| PerAvg | 94.44 ± 0.26 | 64.76 ± 0.13 | 36.27 ± 0.32 | 87.87 ± 0.17 | 93.52 ± 0.32 | 64.46 ± 0.02 | 27.29 ± 0.20 | 84.87 ± 0.01 |
| FedRod | 98.09 ± 0.27 | 88.90 ± 0.19 | 44.33 ± 0.26 | 97.50 ± 0.24 | 96.85 ± 0.11 | 70.52 ± 0.31 | 28.17 ± 0.12 | 96.46 ± 0.22 |
| FedPac | 96.90 ± 0.03 | 87.81 ± 0.17 | 48.83 ± 0.04 | 97.74 ± 0.32 | 94.63 ± 0.22 | 73.02 ± 0.16 | 29.94 ± 0.24 | 94.10 ± 0.09 |
| FedBabu | 96.40 ± 0.08 | 88.19 ± 0.19 | 49.18 ± 0.09 | 96.76 ± 0.31 | 95.10 ± 0.24 | 70.91 ± 0.26 | 28.33 ± 0.23 | 93.11 ± 0.02 |
| FedAS | 97.91 ± 0.22 | **89.15 ± 0.06** | **50.37 ± 0.18** | 97.71 ± 0.15 | 96.78 ± 0.33 | **75.75 ± 0.21** | **32.57 ± 0.31** | 94.53 ± 0.12 |
| FedAvg | 93.77 ± 0.33 | 60.89 ± 0.12 | 28.78 ± 0.10 | 86.45 ± 0.30 | 91.47 ± 0.26 | 59.12 ± 0.21 | 25.55 ± 0.22 | 84.23 ± 0.18 |

## E.1 ADDITIONAL RESULTS

We show the additional performance comparison between our method FedOMG and FL baselines under different data heterogeneity level $\alpha = \{0.1, 0.5\}$ in Table 5. FedOMG+ROD and FedOMG achieve the best and second-best performance across all dataset.

## E.2 PERFORMANCE OF FEDOMG AND BASELINES WITHOUT PRETRAINED MODELS

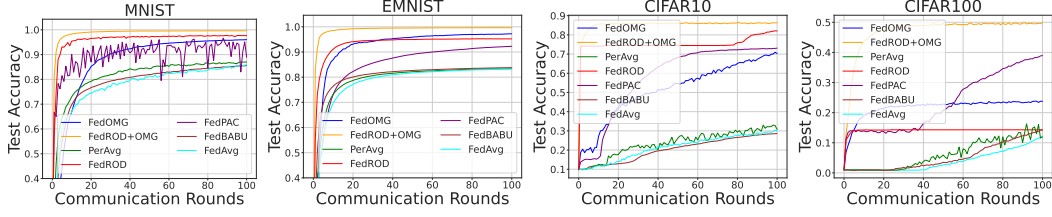

Figure 9: Performance comparison of FL algorithms without pretrained models for $\alpha = 0.1, U = 20$.

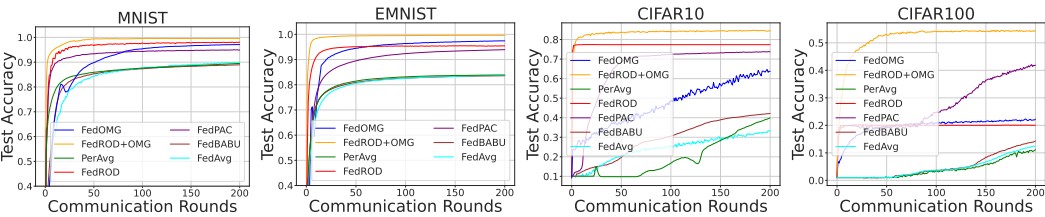

Figure 10: Performance comparison of FL algorithms without pretrained models for $\alpha = 0.1, U = 40$.

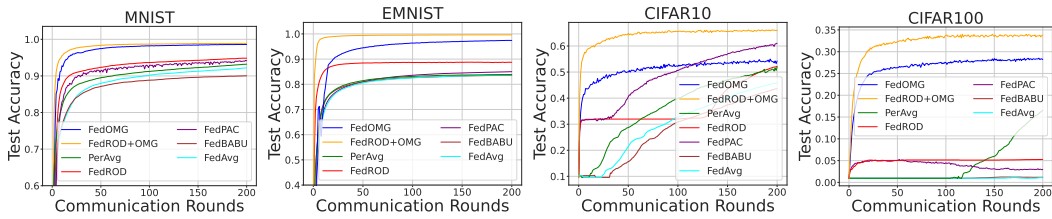

Figure 11: Performance comparison of FL algorithms without pretrained models for $\alpha = 1, U = 40$.

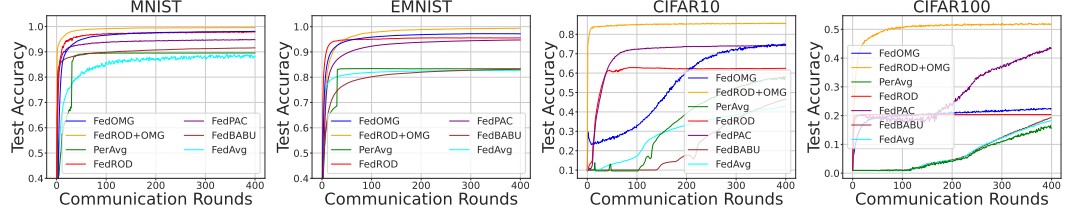

Figure 12: Performance comparison of FL algorithms without pretrained models for $\alpha = 0.1, U = 60$.

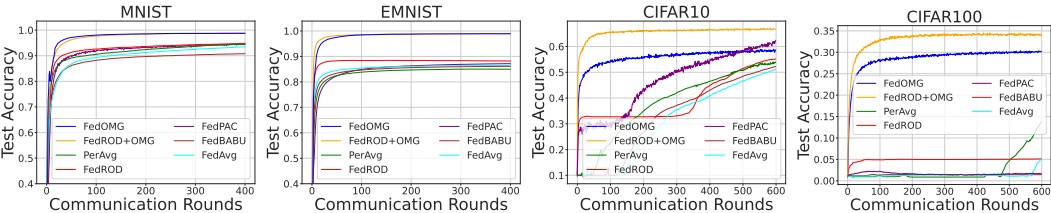

Figure 13: Performance comparison of FL algorithms without pretrained models for $\alpha = 1, U = 60$.

## E.3 DOMAIN PERFORMANCE

Table 6: Performance comparison on VLCS dataset.

| Methods | Backbone | V | L | C | S | Avg |
|---|---|---|---|---|---|---|
| **Centralized Methods** | | | | | | |
| ERM | ResNet-18 | $74.0 \pm 0.6$ | $67.9 \pm 0.7$ | $97.6 \pm 0.3$ | $70.9 \pm 0.2$ | 77.6 |
| Fishr | ResNet-18 | $75.7 \pm 0.3$ | $67.3 \pm 0.5$ | $97.6 \pm 0.7$ | $72.2 \pm 0.9$ | 78.2 |
| Fish | ResNet-18 | $74.4 \pm 0.7$ | $66.5 \pm 0.3$ | $97.6 \pm 0.6$ | $72.7 \pm 0.6$ | 77.8 |
| **Federated Methods** | | | | | | |
| FedSR | ResNet-18 | $72.8 \pm 0.3$ | $62.3 \pm 0.3$ | $93.8 \pm 0.5$ | $74.4 \pm 0.6$ | 75.8 |
| FedGA | ResNet-18 | $74.4 \pm 0.1$ | $56.9 \pm 1.0$ | $94.3 \pm 0.6$ | $68.9 \pm 0.9$ | 73.4 |
| StableFDG | ResNet-18 | $73.6 \pm 0.1$ | $59.2 \pm 0.7$ | $98.1 \pm 0.2$ | $70.2 \pm 1.1$ | 75.3 |
| **FedOMG** | **ResNet-18** | $\mathbf{82.3 \pm 0.5}$ | $\mathbf{67.5 \pm 0.4}$ | $\mathbf{99.3 \pm 0.1}$ | $\mathbf{79.1 \pm 0.5}$ | **82.0** |

Table 7: Performance comparison on PACS dataset.

| Methods | Backbone | P | A | C | S | Avg |
|---|---|---|---|---|---|---|
| **Centralized Methods** | | | | | | |
| ERM | ResNet-18 | $96.2 \pm 0.3$ | $86.5 \pm 1.0$ | $81.3 \pm 0.6$ | $82.7 \pm 1.1$ | 86.7 |
| Fish | ResNet-18 | $97.9 \pm 0.4$ | $87.9 \pm 0.6$ | $80.8 \pm 0.5$ | $81.1 \pm 0.8$ | 86.9 |
| Fishr | ResNet-18 | $95.6 \pm 0.3$ | $84.7 \pm 1.2$ | $81.1 \pm 0.2$ | $80.6 \pm 0.5$ | 85.5 |
| **Federated Methods** | | | | | | |
| FedSR | ResNet-18 | $94.0 \pm 0.6$ | $82.8 \pm 1.5$ | $75.2 \pm 0.5$ | $81.7 \pm 0.8$ | 83.4 |
| FedGA | ResNet-18 | $93.9 \pm 0.2$ | $81.2 \pm 0.7$ | $76.7 \pm 0.4$ | $82.5 \pm 0.1$ | 83.5 |
| StableFDG | ResNet-18 | $94.8 \pm 0.1$ | $83.0 \pm 1.1$ | $79.3 \pm 0.2$ | $79.7 \pm 0.8$ | 84.2 |
| **FedOMG** | **ResNet-18** | $\mathbf{98.0 \pm 0.2}$ | $\mathbf{89.7 \pm 0.4}$ | $\mathbf{81.4 \pm 0.8}$ | $\mathbf{84.3 \pm 0.5}$ | **88.4** |

Table 8: Performance comparison on OfficeHome dataset.

| Methods | Backbone | A | C | P | R | Avg |
|---|---|---|---|---|---|---|
| **Centralized Methods** | | | | | | |
| ERM | ResNet-18 | $61.7 \pm 0.7$ | $53.4 \pm 0.3$ | $74.1 \pm 0.1$ | $76.2 \pm 0.3$ | 66.4 |
| Fish | ResNet-18 | $63.4 \pm 0.8$ | $54.2 \pm 0.3$ | $76.4 \pm 0.3$ | $78.5 \pm 0.2$ | 68.2 |
| Fishr | ResNet-18 | $60.2 \pm 0.5$ | $52.2 \pm 0.6$ | $75.0 \pm 0.2$ | $76.5 \pm 0.3$ | 66.0 |
| **Federated Methods** | | | | | | |
| FedSR | ResNet-18 | $57.9 \pm 0.2$ | $50.3 \pm 0.6$ | $73.3 \pm 0.1$ | $75.5 \pm 0.1$ | 64.3 |
| FedGA | ResNet-18 | $58.5 \pm 0.4$ | $54.3 \pm 0.6$ | $73.3 \pm 0.8$ | $74.7 \pm 1.0$ | 65.2 |
| StableFDG | ResNet-18 | $57.1 \pm 0.3$ | $57.9 \pm 0.5$ | $72.7 \pm 0.6$ | $72.1 \pm 0.8$ | 65.0 |
| **FedOMG** | **ResNet-18** | $\mathbf{65.4 \pm 0.4}$ | $\mathbf{58.1 \pm 0.3}$ | $\mathbf{77.5 \pm 0.4}$ | $\mathbf{78.9 \pm 0.5}$ | **70.0** |

### E.4    EVALUATIONS ON COMPUTATION OVERHEAD

Fig. 14 demonstrates the computation time of FedOMG, FedROD, and FedPAC. The test are evaluated on same number of communication rounds (i.e., $R = 800$). FedOMG shows the most computation efficient on most of settings (e.g., MNIST, EMNIST, CIFAR-10). On CIFAR-100, FedOMG computation time is lower than FedROD. FedPAC tends to consume a significantly higher computation time than that of FedROD and FedOMG (up to 10 times).

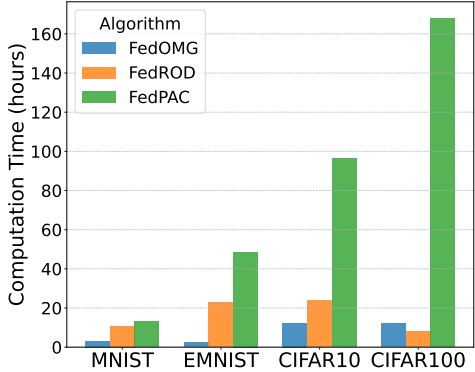

Figure 14: Computation time comparison across different datasets.

### E.5    EVALUATIONS ON FEDERATED DOMAIN GENERALIZATION BENCHMARK

As highlighted in (Bai et al., 2024), the performance of FDG is observed to decline when influenced by the combined effects of domain shift and non-independent and identically distributed (non-IID) data. To investigate this, we conducted experimental evaluations using the Celeb-A dataset, employing a benchmark with two distinct non-IID settings ($\alpha = 0.1, 1$). We compare FedOMG with FedADG (Zhang et al., 2023a), FedSR (Nguyen et al., 2022a), Scaffold (Karimireddy et al., 2020), FedAvg (McMahan et al., 2017) and FedProx (Li et al., 2020) The results of these evaluations are presented in Figure 15.

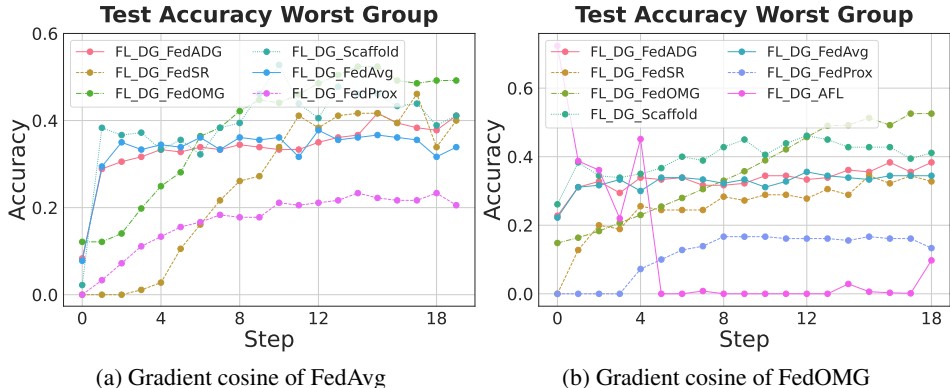

(a) Gradient cosine of FedAvg

(b) Gradient cosine of FedOMG

Figure 15: The evaluations of FDG and FL algorithms on CelebA on two non-IID settings. Fig. 15a uses $\alpha = 0.1$, and Fig. 15b uses $\alpha = 1.0$). We use the worst group test accuracy as a metric to evaluate the algorithms on Celeb-A.

As shown in Figs. 15a and 15b, FedOMG exhibits slower convergence during the initial stages. However, unlike other algorithms that tend to reach saturation quickly, FedOMG demonstrates gradual and consistent improvement over time. This behavior can be attributed to FedOMG's tendency to prioritize identifying a invariant gradient direction, effectively balancing rapid convergence with stable and invariant updates. This approach enhances long-term generalization performance.

# F PROOF ON THEOREMS

## F.1 TECHNICAL ASSUMPTIONS

**Assumption 1 ($L$-smoothness)** *Each local objective function is Lipschitz smooth, that is,*

$$\|\nabla\mathcal{E}(x;\mathcal{D}_u) - \nabla\mathcal{E}(y;\mathcal{D}_u)\| \le L\|\mathcal{E}(x;\mathcal{D}_u) - \mathcal{E}(y;\mathcal{D}_u)\|, \forall u \in \mathcal{U}_\mathcal{S}. \tag{18}$$

**Assumption 2 ($\mu$-strongly convex)** *Each local objective function is Lipschitz smooth, that is,*

$$\|\nabla\mathcal{E}(x;\mathcal{D}_u) - \nabla\mathcal{E}(y;\mathcal{D}_u)\| \ge \mu\|\mathcal{E}(x;\mathcal{D}_u) - \mathcal{E}(y;\mathcal{D}_u)\|, \forall u \in \mathcal{U}_\mathcal{S}. \tag{19}$$

**Assumption 3 (Domain triangle inequality (Zhao et al., 2019))** *For any hypothesis space $\mathcal{H}$, it can be readily verified that $d_\mathcal{H}(\cdot,\cdot)$ satisfies the triangular inequality:*

$$d_{\mathcal{H}\triangle\mathcal{H}}(\mathcal{D}, \mathcal{D}'') \le d_{\mathcal{H}\triangle\mathcal{H}}(\mathcal{D}, \mathcal{D}') + d_{\mathcal{H}\triangle\mathcal{H}}(\mathcal{D}', \mathcal{D}''). \tag{20}$$

## F.2 TECHNICAL LEMMAS

**Lemma 4** *If we have $\mathcal{E}_{\hat{\mathcal{D}}}(\theta) = \sum_{u \in \mathcal{U}_\mathcal{S}} \gamma_u \mathcal{E}_{\hat{\mathcal{D}}_u}$, then for any unseen domain $\mathcal{D}_\mathcal{T}$, we have:*

$$d_{\mathcal{H}\triangle\mathcal{H}}(\mathcal{D}_\mathcal{S}, \mathcal{D}_\mathcal{T}) = \sum_{u \in \mathcal{U}_\mathcal{S}} \gamma_u d_{\mathcal{H}\triangle\mathcal{H}}(\mathcal{D}_u, \mathcal{D}_\mathcal{T}). \tag{21}$$

*Proof.* From the definition of $d_{\mathcal{H}\triangle\mathcal{H}}(\cdot,\cdot)$ in (Arjovsky et al., 2020), we can get

$$
\begin{aligned}
d_{\mathcal{H}\triangle\mathcal{H}}(\mathcal{D}_\mathcal{S}, \mathcal{D}_\mathcal{T}) &= 2 \sup_{A \in \mathcal{A}_{\mathcal{H}\triangle\mathcal{H}}} |\mathrm{Pr}_{\hat{\mathcal{D}}}(A) - \mathrm{Pr}_{\mathcal{D}_\mathcal{T}}(A)| = 2 \sup_{A \in \mathcal{A}_{\mathcal{H}\triangle\mathcal{H}}} \Big| \sum_{u \in \mathcal{U}_\mathcal{S}} \gamma_u \mathrm{Pr}_{\hat{\mathcal{D}}}(A) - \mathrm{Pr}_{\mathcal{D}_\mathcal{T}}(A) \Big| \\
&\le 2 \sup_{A \in \mathcal{A}_{\mathcal{H}\triangle\mathcal{H}}} \Big| \sum_{u \in \mathcal{U}_\mathcal{S}} \gamma_u \Big[ \mathrm{Pr}_{\hat{\mathcal{D}}}(A) - \mathrm{Pr}_{\mathcal{D}_\mathcal{T}}(A) \Big] \Big| \\
&\le 2 \sup_{A \in \mathcal{A}_{\mathcal{H}\triangle\mathcal{H}}} \sum_{u \in \mathcal{U}_\mathcal{S}} \gamma_u |\mathrm{Pr}_{\hat{\mathcal{D}}}(A) - \mathrm{Pr}_{\mathcal{D}_\mathcal{T}}(A)| \\
&\le 2 \sum_{u \in \mathcal{U}_\mathcal{S}} \gamma_u \sup_{A \in \mathcal{A}_{\mathcal{H}\triangle\mathcal{H}}} |\mathrm{Pr}_{\hat{\mathcal{D}}}(A) - \mathrm{Pr}_{\mathcal{D}_\mathcal{T}}(A)| \\
&= \sum_{u \in \mathcal{U}_\mathcal{S}} \gamma_u d_{\mathcal{H}\triangle\mathcal{H}}(\hat{\mathcal{D}}_u, \mathcal{D}_\mathcal{T}).
\end{aligned}
\tag{22}
$$

**Lemma 5** *For any $\theta \in \Theta$, the expectation risk gap between domain $A$ and domain $B$ is bounded by the domain divergence $d_{\mathcal{H}\triangle\mathcal{H}}(A, B)$.*

$$|\mathcal{E}_A(\theta) - \mathcal{E}_B(\theta)| \le \frac{1}{2} d_{\mathcal{H}\triangle\mathcal{H}}(A, B). \tag{23}$$

*Proof.* By the definition of $d_{\mathcal{H}\triangle\mathcal{H}}(\cdot,\cdot)$ in (Arjovsky et al., 2020), we have:

$$d_{\mathcal{H}\triangle\mathcal{H}}(A, B) = 2 \sup_{\theta,\theta' \in \Theta} \Big| \mathrm{Pr}_{x\sim A}[f(x;\theta) \ne f(x;\theta')] - \mathrm{Pr}_{x\sim B}[f(x;\theta) \ne f(x;\theta')] \Big|, \tag{24}$$

where $f(x;\theta)$ means the prediction function on data $x$ with model parameter $\theta$. We chose $\theta'$ as parameter of the label function, then $f(x;\theta) \ne f(x;\theta')$ means the loss function $\mathcal{L}(x;\theta)$, so we have:

$$d_{\mathcal{H}\triangle\mathcal{H}}(A, B) = 2 \sup_{\theta \in \Theta} \Big| \mathrm{Pr}_{x\sim A}[\mathcal{L}(x;\theta)] - \mathrm{Pr}_{x\sim B}[\mathcal{L}(x;\theta)] \Big| \ge 2|\mathcal{E}_A(\theta) - \mathcal{E}_B(\theta)|. \tag{25}$$

Here, $(a)$ holds due to Assumption 1.

**Lemma 6 (Guarantee of source domain invariance)** *If we have $\mathcal{E}_{\hat{\mathcal{D}}}(\theta) = \sum_{u \in \mathcal{U}_S} \gamma_u \mathcal{E}_{\hat{\mathcal{D}}_u}$, then for any domain $\mathcal{D}_S$, we have:*

$$\sum_{u \in \mathcal{U}_S} \gamma_u d_{\mathcal{H} \triangle \mathcal{H}}(\hat{\mathcal{D}}_u, \mathcal{D}_S) \leq \sum_{u \in \mathcal{U}_S} \sum_{v \in \mathcal{U}_S} \gamma_u d_{\mathcal{H} \triangle \mathcal{H}}(\hat{\mathcal{D}}_u, \hat{\mathcal{D}}_v). \tag{26}$$

*Proof.* From the definition of $d_{\mathcal{H} \triangle \mathcal{H}}(\cdot, \cdot)$ in (Arjovsky et al., 2020), we can get

$$\sum_{u \in \mathcal{U}_S} \gamma_u d_{\mathcal{H} \triangle \mathcal{H}}(\hat{\mathcal{D}}_u, \mathcal{D}_S) = 2 \sum_{u \in \mathcal{U}_S} \gamma_u \sup_{A \in \mathcal{A}_{\mathcal{H} \triangle \mathcal{H}}} |\mathrm{Pr}_{\hat{\mathcal{D}}_u}(A) - \mathrm{Pr}_{\mathcal{D}_S}(A)|$$

$$= 2 \sum_{u \in \mathcal{U}_S} \gamma_u \sup_{A \in \mathcal{A}_{\mathcal{H} \triangle \mathcal{H}}} |\mathrm{Pr}_{\hat{\mathcal{D}}_u}(A) - \sum_{v \in \mathcal{U}_S} \gamma_v \mathrm{Pr}_{\hat{\mathcal{D}}_v}(A)|$$

$$= 2 \sum_{u \in \mathcal{U}_S} \gamma_u \sup_{A \in \mathcal{A}_{\mathcal{H} \triangle \mathcal{H}}} |\sum_{v \in \mathcal{U}_S} \gamma_v \mathrm{Pr}_{\hat{\mathcal{D}}_u}(A) - \sum_{v \in \mathcal{U}_S} \gamma_v \mathrm{Pr}_{\hat{\mathcal{D}}_v}(A)|$$

$$\leq 2 \sum_{u \in \mathcal{U}_S} \gamma_u \sum_{v \in \mathcal{U}_S} \gamma_v \sup_{A \in \mathcal{A}_{\mathcal{H} \triangle \mathcal{H}}} |\mathrm{Pr}_{\hat{\mathcal{D}}_u}(A) - \mathrm{Pr}_{\hat{\mathcal{D}}_v}(A)|$$

$$\leq \sum_{u \in \mathcal{U}_S} \sum_{v \in \mathcal{U}_S} \gamma_u d_{\mathcal{H} \triangle \mathcal{H}}(\hat{\mathcal{D}}_u, \hat{\mathcal{D}}_v). \tag{27}$$

### F.3 PROOF ON LEMMA 3

*Proof.* By the definition of $d_{\mathcal{H} \triangle \mathcal{H}}(\cdot, \cdot)$ in (Arjovsky et al., 2020), we have:

$$d_{\mathcal{H} \triangle \mathcal{H}}(A, B) = 2 \sup_{\theta, \theta' \in \Theta} \left| \mathrm{Pr}_{x \sim A}[f(x; \theta) \neq f(x; \theta')] - \mathrm{Pr}_{x \sim B}[f(x; \theta) \neq f(x; \theta')] \right|, \tag{28}$$

where $f(x; \theta)$ means the prediction function on data $x$ with model parameter $\theta$. We chose $\theta'$ as parameter of the label function, then $f(x; \theta) \neq f(x; \theta')$ means the loss function $\mathcal{L}(x; \theta)$, so we have:

$$d_{\mathcal{H} \triangle \mathcal{H}}(A, B) = 2 \sup_{\theta \in \Theta} \left| \mathrm{Pr}_{x \sim A}[\mathcal{L}(x; \theta)] - \mathrm{Pr}_{x \sim B}[\mathcal{L}(x; \theta)] \right|$$

$$= 2 \sup_{\theta \in \Theta} |\mathcal{E}_A(\theta) - \mathcal{E}_B(\theta)|. \overset{(a)}{\leq} \frac{2}{\mu} \sup_{\theta \in \Theta} |\nabla \mathcal{E}_A(\theta) - \nabla \mathcal{E}_B(\theta)| \leq \frac{1}{\mu} d_{\mathcal{G} \circ \theta}(A, B). \tag{29}$$

Here, $d_{\mathcal{G} \circ \theta}(A, B)$ as the gradient divergence, given the model $\theta$ and $(a)$ holds due to Assumption 2.

### F.4 PROOF ON THEOREM 2

Let $\hat{\mathcal{D}}_u$ be the sampled counterpart from the domain $\mathcal{D}_u$, we have $\mathcal{E}_{\hat{\mathcal{D}}_u}$ is an empirical risk of $\mathcal{D}_u$, i.e., $\mathcal{E}_{\hat{\mathcal{D}}_u} = 1/N_u \sum_{i=1}^{N_u} \mathcal{L}(f(x_u^i; \theta), y_u^i)$. We also have expected risk $\mathcal{E}_{\mathcal{D}_u}$ defined as $\mathcal{E}_{\mathcal{D}_u} = \mathbb{E}_{(x,y \in \mathcal{D}_u)}[\mathcal{L}(f(x; \theta), y)]$. For a given $\theta \in \Theta$, with the definition of generalization bound, the following inequality holds with at most $\frac{\delta}{U_S}$ for each domain $\hat{D}_u$ ($U_S$ is the number of users, which is also the number of domains).

$$\mathcal{E}_{\hat{\mathcal{D}}_u}(\theta) - \mathcal{E}_{\mathcal{D}_u}(\theta) > \sqrt{\frac{\log M + \log U_S / \delta}{2N_u}}. \tag{30}$$

Moreover, from Lemma 5, we have $|\mathcal{E}_{\mathcal{D}_u}(\theta) - \mathcal{E}_{\mathcal{D}_T}(\theta)| \leq \frac{1}{2} d_{\mathcal{H} \triangle \mathcal{H}}(\mathcal{D}_u, \mathcal{D}_T)$ for each domain. Then let us consider Eq. (30), we can obtain the following inequalities with the probability at least greater than $1 - \frac{\delta}{U_S}$:

$$\min_{\theta'} \mathcal{E}_{\mathcal{D}_u}(\theta') \leq \mathcal{E}_{\hat{\mathcal{D}}_u}(\theta) + \sqrt{\frac{\log M + \log U_S / \delta}{2N_u}}$$

$$\leq \mathcal{E}_{\mathcal{D}_T}(\theta) + \frac{1}{2} d_{\mathcal{H} \triangle \mathcal{H}}(\hat{\mathcal{D}}_u, \mathcal{D}_T) + \sqrt{\frac{\log M + \log U_S / \delta}{2N_u}}. \tag{31}$$

We denote the local optimal on each client of source set $u$, $u \in \mathcal{U}_S$ as $\theta_u^*$. If we choose a specific parameter $\theta_{\mathcal{T}}^* = \min_\theta \mathcal{E}_{\mathcal{D}_{\mathcal{T}}}(\theta)$ which is the local optimal on the unseen domain $\mathcal{T}$, the above third inequality still holds. Then, we can rewrite the above inequalities into:

$$\mathcal{E}_{\hat{\mathcal{D}}_u}(\theta_u^*) \leq \mathcal{E}_{\mathcal{D}_{\mathcal{T}}}(\theta_u^*) + \frac{1}{2}d_{\mathcal{H}\triangle\mathcal{H}}(\hat{\mathcal{D}}_u, \mathcal{D}_{\mathcal{T}}) + \sqrt{\frac{\log M + \log U_S/\delta}{2N_u}}. \tag{32}$$

Considering on each domain, Eq. (32) holds. By a similar derivation process, we can obtain the inequality between $\mathcal{T}$ and $\hat{\mathcal{D}}$ with the probability at least greater than $1 - \delta$.

$$\sum_{u \in \mathcal{U}_S} \gamma_u \mathcal{E}_{\hat{\mathcal{D}}_u}(\theta_u^*) \leq \mathcal{E}_{\mathcal{D}_{\mathcal{T}}}(\theta_u^*) + \sum_{u \in \mathcal{U}_S} \gamma_u \left[ \frac{1}{2}d_{\mathcal{H}\triangle\mathcal{H}}(\hat{\mathcal{D}}_u, \mathcal{D}_{\mathcal{T}}) + \sqrt{\frac{\log M + \log U_S/\delta}{2N_u}} \right]. \tag{33}$$

Combining the Eq. (33) and Objective 2, we have Theorem 2 with the global model $\theta$ after R rounds FL. For instance,

$$\mathcal{E}_{\mathcal{D}_{\mathcal{T}}}(\theta^R) - \mathcal{E}_{\mathcal{D}_{\mathcal{T}}}(\theta_{\mathcal{D}_{\mathcal{T}}}^*)$$

$$\leq \sum_{u \in \mathcal{U}_S} \gamma_u \left[ \mathcal{E}_{\hat{\mathcal{D}}_u}(\theta) - \mathcal{E}_{\hat{\mathcal{D}}_u}(\theta_u^*) + d_{\mathcal{H}\triangle\mathcal{H}}(\hat{\mathcal{D}}_u, \mathcal{D}_{\mathcal{T}}) + \frac{\sqrt{\log M + \log \frac{1}{\delta}}}{\sqrt{2N_u}} + \frac{\sqrt{\log M + \log \frac{U_S}{\delta}}}{\sqrt{2N_u}} \right] + \zeta^*$$

$$\leq \sum_{u \in \mathcal{U}_S} \gamma_u \left[ \mathcal{E}_{\hat{\mathcal{D}}_u}(\theta) + d_{\mathcal{H}\triangle\mathcal{H}}(\hat{\mathcal{D}}_u, \mathcal{D}_{\mathcal{T}}) + \frac{\sqrt{\log \frac{M}{\delta}} + \sqrt{\log \frac{U_S M}{\delta}}}{\sqrt{2N_u}} \right] + \zeta^*. \tag{34}$$

To further analyze the convergence bound, we consider the Assumption 3. For instance,

$$\mathcal{E}_{\mathcal{D}_{\mathcal{T}}}(\theta^R) - \mathcal{E}_{\mathcal{D}_{\mathcal{T}}}(\theta_{\mathcal{D}_{\mathcal{T}}}^*)$$

$$\leq \sum_{u \in \mathcal{U}_S} \gamma_u \left[ \mathcal{E}_{\hat{\mathcal{D}}_u}(\theta) + d_{\mathcal{H}\triangle\mathcal{H}}(\hat{\mathcal{D}}_u, \mathcal{D}_{\mathcal{T}}) + \frac{\sqrt{\log \frac{M}{\delta}} + \sqrt{\log \frac{U_S M}{\delta}}}{\sqrt{2N_u}} \right] + \zeta^* \tag{35}$$

$$\leq \sum_{u \in \mathcal{U}_S} \gamma_u \left[ \mathcal{E}_{\hat{\mathcal{D}}_u}(\theta) + d_{\mathcal{H}\triangle\mathcal{H}}(\hat{\mathcal{D}}_u, \mathcal{D}_S) + d_{\mathcal{H}\triangle\mathcal{H}}(\mathcal{D}_S, \mathcal{D}_{\mathcal{T}}) + \frac{\sqrt{\log \frac{M}{\delta}} + \sqrt{\log \frac{U_S M}{\delta}}}{\sqrt{2N_u}} \right] + \zeta^*$$

$$\overset{(b)}{\leq} \sum_{u \in \mathcal{U}_S} \gamma_u \left[ \mathcal{E}_{\hat{\mathcal{D}}_u}(\theta) + \sum_{v \in \mathcal{U}_S} \frac{d_{\mathcal{H}\triangle\mathcal{H}}(\hat{\mathcal{D}}_u, \hat{\mathcal{D}}_v)}{\mu} + d_{\mathcal{H}\triangle\mathcal{H}}(\mathcal{D}_S, \mathcal{D}_{\mathcal{T}}) + \frac{\sqrt{\log \frac{M}{\delta}} + \sqrt{\log \frac{U_S M}{\delta}}}{\sqrt{2N_u}} \right] + \zeta^*.$$

We have $(b)$ holds due to Lemma 6. Applying Lemma 3, we have:

$$\mathcal{E}_{\mathcal{D}_{\mathcal{T}}}(\theta^R) - \mathcal{E}_{\mathcal{D}_{\mathcal{T}}}(\theta_{\mathcal{D}_{\mathcal{T}}}^*)$$

$$\leq \sum_{u \in \mathcal{U}_S} \gamma_u \left[ \mathcal{E}_{\hat{\mathcal{D}}_u}(\theta) + \sum_{v \in \mathcal{U}_S} \frac{d_{\mathcal{G} \circ \theta}(\hat{\mathcal{D}}_u, \hat{\mathcal{D}}_v)}{\mu} + d_{\mathcal{H}\triangle\mathcal{H}}(\mathcal{D}_S, \mathcal{D}_{\mathcal{T}}) + \frac{\sqrt{\log \frac{M}{\delta}} + \sqrt{\log \frac{U_S M}{\delta}}}{\sqrt{2N_u}} \right] + \zeta^*.$$

### F.5 PROOF ON THEOREM 1

To find the optimal solution of invariant gradient direction $g_{\text{IGD}}^{(r)}$ in Eq. (12), we consider $x = g_{\text{IGD}}^{(r)}$ and consider the maximization problem with $x$ as optimization variable. Denote $\phi = \kappa^2 \|g_{\text{FL}}^{(r)}\|^2$. Note that $\min_u \langle g_u^{(r)}, g_{\text{FL}}^{(r)} \rangle = \min_\gamma \langle \sum_u \gamma_u h_u^{(r)}, h_g^{(r)} \rangle$. The Lagrangian of the objective is

$$\max_x \min_{\lambda, \gamma} \Big( \sum_{u \in \mathcal{U}_S} \gamma_u h_u^{(r)} \Big)^\top x - \frac{\lambda}{2} \|g_{\text{FL}}^{(r)} - x\|^2 + \frac{\lambda}{2} \phi, \text{ s.t. } \lambda \geq 0. \tag{36}$$

Since the problem is a convex programming and and Slater's condition is satisfied for $\kappa > 0$ (meanwhile, if $\kappa = 0$, it can be easily verified that all results hold trivially), the strong duality holds. Consequently, the order of the $\min$ and $\max$ operations can be interchanged. For instance,

$$\min_{\lambda, \gamma} \max_x \underbrace{\Big( \sum_{u \in \mathcal{U}_S} \gamma_u h_u^{(r)} \Big)^\top x - \frac{\lambda}{2} \|g_{\text{FL}}^{(r)} - x\|^2 + \frac{\lambda}{2} \phi}_{A_1}, \text{ s.t. } \lambda \geq 0. \tag{37}$$

Taking $A_1$ into consideration. If we consider $\lambda, \gamma$ as constant, $x$ is the variable, $x$ achieves the optimal solution when $\partial A_1 / \partial x = 0$. Another speaking, we have $\lambda(x - g_{\text{FL}}^{(r)}) - \sum_{u=1}^U \gamma_u h_u^{(r)} = 0$ or specifically,

$$x = g_{\text{FL}}^{(r)} + \Big( \sum_{u=1}^U \gamma_u h_u^{(r)} \Big) / \lambda. \tag{38}$$

Therefore, we have the followings:

$$A_1 = \Big( \sum_{u=1}^U \gamma_u h_u^{(r)} \Big)^\top \Big( g_{\text{FL}}^{(r)} + \Big( \sum_{u=1}^U \gamma_u h_u^{(r)} \Big) / \lambda \Big) - \frac{\lambda}{2} \| g_{\text{FL}}^{(r)} - \Big( g_{\text{FL}}^{(r)} + \Big( \sum_{u=1}^U \gamma_u h_u^{(r)} \Big) / \lambda \Big) \|^2 + \frac{\lambda}{2} \phi$$

$$= \Big( \sum_{u=1}^U \gamma_u h_u^{(r)} \Big)^\top \Big( g_{\text{FL}}^{(r)} + \Big( \sum_{u=1}^U \gamma_u h_u^{(r)} \Big) / \lambda \Big) - \frac{\lambda}{2} \| \frac{1}{\lambda} \sum_{u=1}^U \gamma_u h_u^{(r)} \|^2 + \frac{\lambda}{2} \phi. \tag{39}$$

Substituting $g_\Gamma^{(r)} = \sum_{u=1}^U \gamma_u h_u^{(r)}$. Consider the optimization problem of Eq. (39), we have:

$$A_1 = g_\Gamma^{(r)\top} \Big( g_{\text{FL}}^{(r)} + g_\Gamma^{(r)} / \lambda \Big) - \frac{\lambda}{2} \| g_\Gamma^{(r)} / \lambda \|^2 + \frac{\lambda}{2} \phi$$

$$= g_\Gamma^{(r)\top} g_{\text{FL}}^{(r)} + \frac{1}{\lambda} g_\Gamma^{(r)\top} g_\Gamma^{(r)} - \frac{1}{2\lambda} \| g_\Gamma^{(r)} \|^2 + \frac{\lambda}{2} \phi$$

$$= g_\Gamma^{(r)\top} g_{\text{FL}}^{(r)} + \frac{1}{2\lambda} \| g_\Gamma^{(r)} \|^2 + \frac{\lambda}{2} \phi. \tag{40}$$

Therefore, we have Eq. 37 is equivalent to

$$\min_{\lambda, \gamma} \underbrace{g_\Gamma^{(r)\top} g_{\text{FL}}^{(r)} + \frac{1}{2\lambda} \| g_\Gamma^{(r)} \|^2 + \frac{\lambda}{2} \phi}_{A_2}. \tag{41}$$

Next, we consider $\lambda$ as variable to find the optimal value. Subsequently, optimization problem Eq. (41) is equivalent to the following relationship:

$$\frac{\partial}{\partial \lambda} A_2 = -\frac{1}{2\lambda^2} \| g_\Gamma^{(r)} \|^2 + \frac{1}{2} \phi = 0. \tag{42}$$

Therefore, the equation achieves the optimality as $\lambda = \| g_\Gamma^{(r)} \| / \phi^{1/2}$. Combining with Eq. (41) and Eq. (38), we have the followings:

$$g_{\text{IGD}}^{(r)} = g_{\text{FL}}^{(r)} + \frac{\kappa \| g_{\text{FL}}^{(r)} \|}{\| g_{\Gamma^*}^{(r)} \|} g_{\Gamma^*}^{(r)} \quad \text{s.t.} \quad \Gamma^* = \arg \min_\Gamma g_\Gamma^{(r)} \cdot g_{\text{FL}}^{(r)} + \kappa \| g_{\text{FL}}^{(r)} \| \| g_\Gamma^{(r)} \|. \tag{43}$$

This solves the problem.

