# OpenReview forum: "Federated Domain Generalization with Data-free On-server Matching Gradient"
_ICLR.cc/2025/Conference — ICLR 2025 Poster_

### Official Review · Reviewer_cQAL · 2024-10-23

**Soundness:** 3
**Presentation:** 2
**Contribution:** 3
**Rating:** 6
**Confidence:** 4

**Summary:**

This paper introduces FedOMG, a method that efficiently leverages domain information from distributed domains to prove the performance of the federated domain generalization (FDG) problem. The authors propose an approach that finds an invariant gradient direction across all domains through gradient inner product maximization to achieve this. The authors take the FDG problem as a multi-objective optimization problem and optimize it by finding the Pareto front. The extensive experiment results show the strengths and effectiveness of FedOMG, which is both superior in performance and computationally efficient.

**Strengths:**

- The idea is presented with solid theory, which is generally easy to follow.
-  Framing the FDG problem as a multi-objective optimization problem using gradient matching is interesting and intuitive. Also, it aligns with the nature of Federated learning with optimization goals.
- The results are strong across many datasets and baselines, and FedOMG is compatible with other related baselines. Many ablation studies are presented to show the effectiveness of the proposed method.

**Weaknesses:**

- **The presentation of the Introduction and Related work**: The authors mention some previous work and I think it would be good to discuss their differences, limitations, and connections, compared with FedOMG. Also, in the introduction, the authors can discuss more on the novelty and design intuitions/details of FedOMG. Now it reads only mentioning gradient matching, which is too general from my perspective. Presentation-wise, this paper can be improved greatly in my opinion.

- **About theory**: In Objective 2, Theorem 2, and Corollary 3, some variables are not defined, such as µ, M, and δ. Also, it would be good if the authors could provide more explanations on how to derive Corollary 2.

- **Toy tasks**: The figure shows too many trajectories and is a little bit confusing. The authors can think of ways to present it conveying the conclusions clearer and more straightforward.

**Questions:**

- Could the authors provide more intuitions behind designing the search space (why search in a ball and use L2 norm)? Also, why are you using FedAvg for g_FL here? Would other federated aggregation rules for non-iid data work better here?
- Also, the authors use a convex combination of local gradients to find the invariant gradient direction. Could you also provide more explanations and intuitions for the design behind that?

---

> ### Author Response · Authors · 2024-11-19
> **Response to Reviewer cQAL 's Weaknesses**
>
> > **Weakness 1:** The presentation of the Introduction and Related work: The authors mention some previous work and I think it would be good to discuss their differences, limitations, and connections, compared with FedOMG. Also, in the introduction, the authors can discuss more on the novelty and design intuitions/details of FedOMG. Now it reads only mentioning gradient matching, which is too general from my perspective. Presentation-wise, this paper can be improved greatly in my opinion.
>
> We thank the Reviewer for the constructive comment.
> According to the limitations of our FDG works, we have implied in the open questions for current FDG approaches.
> Per the Reviewer comment, we have revised the introduction and related works accordingly to improve the clarity and motivations of our proposed FedOMG.
>
> > **Weakness 2:** About theory: In Objective 2, Theorem 2, and Corollary 3, some variables are not defined, such as $\mu$, $M$, and $\delta$. Also, it would be good if the authors could provide more explanations on how to derive Corollary 2.
>
> We thank the Reviewer for this constructive comment. We have provided notation definitions in the paper to improve the paper clarity. We have also removed Corollary 2 (in the original manuscript) as it is redundant in the Section 4 and too ambiguous.
> $M$ is already defined in the notation in Section 2, as the number of model parameters.
>
> > **Weakness 3:** Toy tasks: The figure shows too many trajectories and is a little bit confusing. The authors can think of ways to present it conveying the conclusions clearer and more straightforward.
>
> We thank the Reviewer for the suggestion and we will revise it.

---

> ### Author Response · Authors · 2024-11-19
> **Response to Reviewer cQAL 's Questions**
>
> > **Question 1:** Could the authors provide more intuitions behind designing the search space (why search in a ball and use L2 norm)? Also, why are you using FedAvg for g_FL here? Would other federated aggregation rules for non-iid data work better here?
>
> The rationale for designing the search space is to constrain the range of the optimized gradient, thereby reducing computational overhead. Additionally, we limit the search space to avoid making it too large, which could lead to overfitting in the optimized gradient, as the optimization process may not be completed effectively within a broader scope.
>
> We use FedAvg as the center of the search space because it is a simple and widely used algorithm in federated learning. Furthermore, as demonstrated in the paper, our method can integrate with other FL algorithms, showing superior performance in both personalization and generalization.
>
> > **Question 2:** Also, the authors use a convex combination of local gradients to find the invariant gradient direction. Could you also provide more explanations and intuitions for the design behind that?
>
> In Eq. (6b), the optimization variable set is $\theta$, which consists of $M$ parameters. This approach suffers from the following issues:
> - The number of data is limited due to the utilization of local gradients as data.
> - The number of optimization variable is large.
> - The direct utilization of inner products may induce a huge computation overheads (as they requires the second-order derivative of the model’s parameters due to the gradient inner product term (see Section 3.4 of $[\textrm{R1}]$)).
>
> Two first issues lead to a prone to overfitting. In our work, we propose a different approach. Instead of optimizing a set of $M$ parameters, we use indirect optimization variable set $\Gamma$. Intuitively, the $\Gamma$ represents the contribution of local gradients to the gradient aggregation and has $U_\mathcal{S}$ number of parameters.
> By optimizing $\Gamma$, we are trying to find the contributions of local gradients to the joint aggregation.
> It is obvious that $M\ll U_\mathcal{S}$, where $M$ are usually very large, e.g., $M=31.7\times 1\textrm{e}6$, while $U_\mathcal{S}  < 1000$ in practice.
>
> According to the third issue, by introducing the indirect optimization variables instead of optimizing directly the model parameters, we can reduce the a huge computation overheads (as direct optimization requires the second-order derivative of the model’s parameters due to the gradient inner product term (see Section 3.4 of $[\textrm{R1}]$).
>
> Per the Reviewer's constructive comment, we have revised the Section 4.2 significantly to improve the clarity and emphasize the contributions of our paper.
>
> - [R1] Alexandre Rame et al., Fishr: Invariant Gradient Variances for Out-of-Distribution Generalization, ICML 2022.

---

> ### Author Response · Authors · 2024-11-21
> **Any follow-up question?**
>
> Dear Reviewer cQAL,
>
> We sincerely appreciate your efforts and time for the community. As we approach the close of the author-reviewer discussion period in one week, we wonder whether the reviewer is satisfied with our response. It will be pleasurable if the reviewer can give us the reviewer's thoughts on the current revision to give us an extra valuable chance to improve our paper. We summarized our revision in the "Revision summary" comment.
>
> Again, we thank the reviewer's valuable commitment and their help to strengthen our submission. We will address all the raised concerns by reviewers if there remain any.

---

> > ### Comment · Reviewer_cQAL · 2024-11-21
> > **Thanks for the response**
> >
> > Thanks for the authors' response and the update on the paper. This solves most of my concerns, and I have raised my score accordingly.

---

> > > ### Author Response · Authors · 2024-11-24
> > >
> > > Dear Reviewer cQAL,
> > >
> > > We would like to express our deepest gratitude for your constructive feedback and the goodwill you've shown towards our work.
> > >
> > > Thank you for your response. Your comments are truly inspiring and have provided invaluable guidance for improving our paper.
> > >
> > > Best regards,
> > >
> > > Authors

---

### Official Review · Reviewer_HBia · 2024-10-28

**Soundness:** 3
**Presentation:** 2
**Contribution:** 2
**Rating:** 5
**Confidence:** 3

**Summary:**

This paper introduces FedOMG to address the challenge of Federated Domain Generalization. The core idea behind FedOMG is to leverage local gradients from distributed models as domain-specific information. FedOMG maximizes the inner product of gradients to ensure that the model finds a gradient direction that is invariant across all domains. Extensive experiments conducted to evaluate the effectiness of the proposed model under federated setup.

**Strengths:**

Theoretical analysis propoded which seems to be true.

**Weaknesses:**

1. The approach to invariant gradient direction through convex optimization and Pareto optimality increases the computational complexity.

2. The method’s effectiveness relies on the assumption that domain gradients will align under the invariant gradient matching, which may not hold well in highly heterogeneous data settings.

3. The connection between the stated motivations (privacy, communication efficiency, and domain generalization) and the chosen methodology (on-server gradient matching and invariant gradient direction) is not clear.

**Questions:**

1. How does on-server gradient alignment directly contribute to reducing domain-specific biases without compromising the generalization capabilities of the global model?

2. Could you clarify how using gradients alone (instead of data) ensures privacy in FDG? Are there specific privacy-preserving guarantees that FedOMG provides through gradient-only use? What limitations, if any, exist in relying solely on gradient matching for privacy preservation, especially with highly heterogeneous client data?

3. What motivated the choice of complex optimization techniques like convex optimization and Pareto optimality in this context? How do these methods specifically enhance FedOMG’s performance over simpler approaches? Could simpler optimization approaches potentially achieve similar outcomes, or are the proposed techniques essential to achieving the method's goals?

4. Could you provide a more intuitive explanation of how gradient direction alignment helps achieve domain invariance across clients?

---

> ### Author Response · Authors · 2024-11-19
> **Response to Reviewer HBia 's Weaknesses Part 1**
>
> > **Weakness 1:** The approach to invariant gradient direction through convex optimization and Pareto optimality increases the computational complexity
>
> We apologize to the Reviewer for not providing a detailed explanation of the experimental settings related to FedOMG's server-side operations, which may have caused confusion. Here, we aim to clarify FedOMG’s computational efficiency and provide further insights into our on-server training settings.
>
> Since the additional computations in FedOMG are performed on the server side, where high computational resources are typically available, our approach capitalizes on these resources to enhance federated learning performance. This design is particularly beneficial in scenarios where local clients are resource-constrained, such as Internet of Things and mobile communications, a critical aspect that is often underutilized in existing FL methods.
>
> In our experimental evaluations, we accounted for this issue, as detailed in Appendix D.4. The results demonstrate that FedOMG achieves significant computational efficiency compared to several state-of-the-art (SOTA) FL methods, such as FedROD and FedPAC.
>
> To further elucidate FedOMG’s computational efficiency, we provide the following details regarding our on-server training settings:
> - Data Size: The input data on the server consists of client gradient vectors, resulting in a small dataset (e.g., 100 data points for a 100-user setting). This is significantly smaller than common datasets like MNIST, which contains 60,000 data points.
> - Optimization Variables: When modeling the optimization problem as a shallow neural network, the network contains only 100 parameters. This lightweight model ensures that our approach does not impose substantial computational demands.
> - Iterations: Our experiments utilized 21 iterations for the on-server optimization, which was sufficient to achieve the reported performance. This low number of iterations minimizes computational overhead on the server.
>
> To address the Reviewer’s concerns, we have incorporated additional details about the on-server training hyperparameters into the revised manuscript. We believe this clarification will help reduce misunderstandings and provide a clearer understanding of FedOMG’s computational efficiency.
>
> > **Weakness 2:** The method’s effectiveness relies on the assumption that domain gradients will align under the invariant gradient matching, which may not hold well in highly heterogeneous.
>
> We respectfully disagree with the Reviewer's comment. The assumption that domain gradients will align under invariant gradient matching is well established and has been thoroughly evaluated and verified in various studies (e.g., [R1], [R2]). Moreover, gradient matching methods consistently demonstrate top performance in current domain generalization benchmarks (i.e., DomainBed [R3]).
>
> Furthermore, we already evaluated the performance of our proposed FedOMG in heterogeneous settings and demonstrated in Tables 1 and 5. The results prove that our FedOMG achieves significant challenging performance across other baselines
> - [R1] Alexandre Rame et al., Fishr: Invariant Gradient Variances for Out-of-Distribution Generalization, ICML 2022.
> - [R2] Yuge Shi et al., Gradient Matching for Domain Generalization, ICLR 2022.
> - [R3] https://github.com/facebookresearch/DomainBed

---

> ### Author Response · Authors · 2024-11-19
> **Response to Reviewer HBia 's Weaknesses Part 2**
>
> > **Weakness 3:** The connection between the stated motivations (privacy, communication efficiency, and domain generalization) and the chosen methodology (on-server gradient matching and invariant gradient direction) is not clear.
>
> Some existing approaches may involve extensive data sharing among users $[\textrm{R1}]$ or the transmission of data to a central server for further processing $[\textrm{R2}], [\textrm{R3}], [\textrm{R4}]$. These methods, however, introduce privacy concerns and communication overhead. In contrast, our proposed method, FedOMG, achieves on-server training without requiring any additional information sharing from users. As a result, it avoids potential issues related to privacy and communication efficiency. Nevertheless, it is important to clarify that privacy robustness is not a primary focus of our work; instead, we concentrate on addressing challenges in domain generalization. In the introduction, we mention the privacy and communication efficiency to emphasize that, our methods do not bring any concerns although we apply an on-server training approach, which is unusual in FL.
>
> According to the risk of attacks and eavesdropping on model parameters, as discussed in Section 4.2, our algorithm can seamlessly integrate with existing FL methods. As a result, our FDG approach inherits both privacy protection and communication efficiency capabilities of robust FL algorithms.
>
> Additionally, we have shown that our on-server computations are efficient, requiring minimal data and simply reusing transmitted models to compute gradients. Overall, we believe our methodology is well-aligned with communication efficiency, privacy and domain generalization objectives in the FL framework.
> - $[\textrm{R1}]$ Canh et al.,  A New Look and Convergence Rate of Federated Multi-Task Learning with Laplacian Regularization, IEEE TNNLS 2022.
> - $[\textrm{R2}]$ Huang et al., Fusion of Global and Local Knowledge for Personalized Federated Learning, TMLR 2023.
> - $[\textrm{R3}]$ Psaltis et al., FedLID: Self-Supervised Federated Learning for Leveraging Limited Image Data, ICCV 2023.
> - $[\textrm{R4}]$ Xu et al., Enhancing Federated Learning With Server-Side Unlabeled Data by Adaptive Client and Data Selection, IEEE TMC 2024.

---

> ### Author Response · Authors · 2024-11-19
> **Response to Reviewer HBia 's Questions Part 1**
>
> > **Question 1:** How does on-server gradient alignment directly contribute to reducing domain-specific biases without compromising the generalization capabilities of the global model?
>
> The gradient matching method is widely theoretically proven to be one of the most robust methods in both reducing domain-specific biases and compromising the generalization capabilities of the model at the same time recently [R1], [R2]. Furthermore, the robust performance of gradient matching over other DG approaches has been experimentally evaluated comprehensively in the DomainBed benchmarks released by Meta [R3].
>
> Given the gradient matching method, implementing gradient matching on the server side enables the efficient aggregation of knowledge from clients without incurring additional computational overhead.
> - [R1] Alexandre Rame et al., Fishr: Invariant Gradient Variances for Out-of-Distribution Generalization, ICML 2022.
> - [R2] Yuge Shi et al., Gradient Matching for Domain Generalization, ICLR 2022.
> - [R3] https://github.com/facebookresearch/DomainBed
>
> > **Question 2:** Could you clarify how using gradients alone (instead of data) ensures privacy in FDG? Are there specific privacy-preserving guarantees that FedOMG provides through gradient-only use? What limitations, if any, exist in relying solely on gradient matching for privacy preservation, especially with highly heterogeneous client data?
>
> As our response to your Weakness 3 in Section 4.2, our algorithm can integrate with other FL algorithms. Consequently, our FDG technique inherits the communication efficiency and privacy robustness FL methods (e.g., model sparsification or quantization). Furthermore, we have demonstrated that our on-server computations do not require extensive data and simply reuse the transmitted models to compute gradients. Overall, we believe that our chosen methodology aligns with communication efficiency in the FL context.
>
> Furthermore, we want to emphasize that our paper motivations are to improving the domain generalization in federated settings. We believe that proving the privacy guarantees in our work is redundant and out of the topic of the paper.
>
> > **Question 3:** What motivated the choice of complex optimization techniques like convex optimization and Pareto optimality in this context? How do these methods specifically enhance FedOMG’s performance over simpler approaches? Could simpler optimization approaches potentially achieve similar outcomes, or are the proposed techniques essential to achieving the method's goals?
>
> We assert that our proposed method successfully achieves its the method's goals, particularly in terms of computational efficiency and hyper-parameter tuning efficiency. In this section, we aim to discuss two key reasons for employing convex optimization, which also contribute to the robustness of our method compared to simpler optimization approaches:
> - To simplify the optimization problem presented in Eq. (12), we propose relaxing the requirement of looping over $U$ clients to compute the loss function in Eq. (12b). This relaxation enhances the computational efficiency of the on-server training process.
> - To reduce the dependency on selecting the hyperparameter $\gamma$ in Eq. (12b), we aim to simplify the hyper-parameter tuning process. This adjustment streamlines the use of FedOMG by minimizing the need for extensive hyper-parameter optimization.
>
> > **Question 4:** Could you provide a more intuitive explanation of how gradient direction alignment helps achieve domain invariance across clients?
>
> The invariant gradient direction inherits the motivation from gradient based multi-task learning. To be more specific, when two gradients form an obtuse angle, the gradient update of one task will cause negative transfer to another task $[\textrm{R1}]$. This phenomenon can be explained in terms of geometry. Specifically, when the angle of two gradients $g_1, g_2$ is obtuse, a specific gradient $g_2$ can be decomposed into two components $g^{\top}_2, g^{\parallel}_2$. $g^{\top}_2$ is orthogonal to $g_1$, thus, only focuses on helping $g_1$ to find an optimal trajectory. $g^{\parallel}_2$ is the antiparallel vector of $g_1$, thus, affects badly to the gradient progress of $g_1$.
>
> Motivated from the aforementioned gradient conflicts in multi-task learning, the invariant gradient direction states that, if the two gradients hold an angle less than $90$ degree, the two gradients will progress towards a directions, which holds good performance on both domains. As a consequence, we can achieve domain-invariant representation with gradient direction alignment.
>
> - $[\textrm{R1}]$ Adrian Javaloy et al., RotoGrad, Gradient Homogenization in Multi-task Learning, ICLR 2022.

---

> ### Author Response · Authors · 2024-11-21
> **Any follow-up question?**
>
> Dear Reviewer HBia,
>
> We sincerely appreciate your efforts and time for the community. As we approach the close of the author-reviewer discussion period in one week, we wonder whether the reviewer is satisfied with our response. It will be pleasurable if the reviewer can give us the reviewer's thoughts on the current revision to give us an extra valuable chance to improve our paper. We summarized our revision in the "Revision summary" comment.
>
> Again, we thank the reviewer's valuable commitment and their help to strengthen our submission. We will address all the raised concerns by reviewers if there remain any.

---

> > ### Comment · Reviewer_HBia · 2024-11-22
> > **Review Update**
> >
> > Dear Authors,
> >
> > Thank you for providing the clarification. I have reviewed it, and I have no further comments. I will keep my score as is.

---

> > > ### Author Response · Authors · 2024-11-24
> > >
> > > Dear Reviewer HBia,
> > >
> > > We truly thank the Reviewer for taking time to review our paper and give some important feedback so that we can improve our paper clarification.
> > >
> > > It is with sincere hope that our responses and corrections have satisfactorily resolved the issues you raised. Should there be any further questions or clarifications you require, please do not hesitate to contact us directly. We are more than willing to engage in further discussions to enhance the quality of our work to your satisfaction.
> > >
> > > Best regards,
> > >
> > > Authors

---

### Official Review · Reviewer_9XVR · 2024-10-30

**Soundness:** 3
**Presentation:** 3
**Contribution:** 2
**Rating:** 6
**Confidence:** 4

**Summary:**

This study introduces Federated Learning via On-server Matching Gradient (FedOMG) for Federated Domain Generalization (FDG). Unlike traditional methods, FedOMG leverages local gradients to identify an invariant gradient direction across domains, enabling efficient feature aggregation on a central server without extra communication costs. It is also compatible with existing FL/FDG methods for enhanced performance.

**Strengths:**

1.	This paper propose a method that is effective and highly compatible with existing FL algorithms and a large number of valid experiments have been conducted on the empirical side.
2.	Although I'm not an expert in this area (FL combined with DG), I think it's a very interesting work, benefiting from the theoretical explanation of how the authors derive the final method step by step, and the theoretical analysis seems solid enough.

**Weaknesses:**

1. What confuses me is the explanation of the motivation for this new approach (Section 3.2), and I would appreciate it if the authors could explain this part in more detail.
2. In lines 203-215, the authors seem to consider an alternative gradient for solving the $M$ -dimensional optimization, and I would like to know if this approach is a rough estimate, and if so, can you discuss the implications in detail
3. a formulation “any FL algorithm” was used in line 267 - 269, I would have expected the authors to mention a great deal of relevant work here to demonstrate this overly certain claim

**Questions:**

See weaknesses.

---

> ### Author Response · Authors · 2024-11-19
> **Response to Reviewer 9XVR 's Weaknesses**
>
> > **Weakness 1:** What confuses me is the explanation of the motivation for this new approach (Section 3.2), and I would appreciate it if the authors could explain this part in more detail.
>
> In Section 3.2, we consider the Invariant Gradient Direction (IGD) rationale, which enables the utilization of local gradients as training data for the on-server optimization. IGD has proven significant robustness in domain generalization, such as Fish [R1] and Fishr [R2], which currently dominates the benchmark of DomainBed. However, due to the following issues, current IGD-based DG approaches (e.g., Fish, Fishr) proved to be unsuitable for IGD.
> - According to Fishr, the joint optimization problem, in which gradient divergence minimization serves as a regularization technique, presents a significant challenge. Specifically, it is not feasible to determine the optimal gradient direction on the server side due to the requirement for direct access to the underlying data.
> - As mentioned by the authors of Fish, the direct utilization of inner products may induce a huge computation overheads (as they requires the second-order derivative of the model’s parameters due to the gradient inner product term (see Section 3.4 of [R1])). This phenomenon also holds in Fish, where the gradient divergence minimization is applied as a regularizer.
> - To deal with the first issue, authors of Fish introduce an indirect applying continuously model update like Reptile. This approach is infeasible in Federated settings, as the models are required to be transmitted among clients continuously, thus, inducing significant communication overheads.
>
> We have provided the discussion in Section 3.2 and revised the Introduction to clarify the motivation.
> - [R1] Yuge Shi et al., Domain Generalization via Gradient Matching, ICLR 2022.
> - [R2] Alexandre Rame et al., Fishr: Invariant Gradient Variances for Out-of-Distribution Generalization, ICML 2022.
>
> > **Weakness 2:** In lines 203-215, the authors seem to consider an alternative gradient for solving the M-dimensional optimization, and I would like to know if this approach is a rough estimate, and if so, can you discuss the implications in detail.
>
> The optimization problem of the FGD update is presented in Eq. (6). The existing approaches to optimize the variable set $\theta$ suffer from the following issues:
> - The amount of data is limited due to the utilization of local gradients as data.
> - The number of optimization variables (i.e., $\theta\in\mathbb{R}^{M}$) is large, e.g., $M=31.7\times 1\textrm{e}6$.
> These two issues lead to a prone to overfitting. In our work, we propose a different approach. In particular, instead of optimizing a set of $M$ parameters, we use indirect optimization variable set $\Gamma$. Intuitively, $\Gamma$ represents the contribution of local gradients to the gradient aggregation and has $U_\mathcal{S}$ number of parameters. It is obvious that $M\ll U_\mathcal{S}$ as normally, federated system has $U_\mathcal{S} < 1000$ in practice.
>
> Furthermore, by introducing the indirect optimization variables instead of direct model parameters, we can hugely reduce computation overheads (as direct optimization requires the second-order derivative of the model’s parameters due to the gradient inner product term (see Section 3.4 of [R1]).
>
> Per the Reviewer's constructive comment, we have revised the Section 4.2 significantly to improve the clarity and emphasize the contributions of our paper.
> - [R1] Yuge Shi et al., Domain Generalization via Gradient Matching, ICLR 2022.
>
> > **Weakness 3:** A formulation “any FL algorithm” was used in line 267 - 269, I would have expected the authors to mention a great deal of relevant work here to demonstrate this overly certain claim.
>
> We appreciate the Reviewer’s feedback regarding the strength of our claims and will adjust the writing accordingly to present a more measured stance. Nonetheless, we would like to clarify that our approach allows for the integration of other FL algorithms into FedOMG by using the gradient from the FL algorithm as a reference. This enables FedOMG to explore an optimal gradient direction relative to the integrated FL algorithm. Notably, the advancements of our FedOMG has been experimented through its integration with various FL and FDG methods, with the results presented in Tables 1 and 2.

---

> ### Author Response · Authors · 2024-11-21
> **Any follow-up question?**
>
> Dear Reviewer 9XVR,
>
> We sincerely appreciate your efforts and time for the community. As we approach the close of the author-reviewer discussion period in one week, we wonder whether the reviewer is satisfied with our response. It will be pleasurable if the reviewer can give us the reviewer's thoughts on the current revision to give us an extra valuable chance to improve our paper. We summarized our revision in the "Revision summary" comment.
>
> Again, we thank the reviewer's valuable commitment and their help to strengthen our submission. We will address all the raised concerns by reviewers if there remain any.

---

> > ### Author Response · Authors · 2024-11-24
> > **The rebuttal deadline is coming soon.**
> >
> > Dear Reviewer 9XVR,
> >
> > As the rebuttal deadline approaches, we would like to express our gratitude for your constructive feedback, which has been instrumental in significantly improving our manuscript. We deeply value your insights and would greatly appreciate any additional feedback or further questions you may have regarding the revisions. We firmly believe that your comments are pivotal for enhancing the quality of our work both in this manuscript and in our future research endeavors.
> >
> > Sincerely,
> >
> > Authors

---

### Official Review · Reviewer_XXzG · 2024-11-02

**Soundness:** 2
**Presentation:** 2
**Contribution:** 3
**Rating:** 6
**Confidence:** 4

**Summary:**

The paper proposes a gradient-matching strategy on the server for federated domain generalization (FDG) using a meta-learning approach.
This essentially boils down to a convex combination update rule instead of a simple average as in FedAvg.
In practice, the optimal combination of local updates is found by bounding the optimization near a simpler update rule like FedAvg update rule.
The paper provides some theoretic results and compares to some baseline methods in several datasets showing performance improvement empirically and that the method can be combined with other methods.

**Strengths:**

- Proposes an new aggregation method to improve federated domain generalization based on gradient matching.

- Provides some theoretic analysis on the proposed method.

- The empirical results show a significant improvement over baselines used and demonstrate that FedOMG can be combined with other methods for a good performance boost (caveat with not comparing to a known Federated DG benchmark paper, see Weaknesses).

**Weaknesses:**

- The paper does not compare to the federated domain generalization benchmark [Bai et al. 2024] that exactly fits this setting. This benchmark paper enables comparison of your method to multiple prior DG methods adapted from central methods and federated DG methods. In particular, Bai et al. [2024] noticed that simple federated averaging or convergence-specific FL methods could beat most prior Federated DG methods. Bai et al. [2024] also control for hyperparameter tuning costs. I would like to see this method applied within this framework to compare to the methods in that benchmark. This will place FedOMG on the same playing field as prior methods and would only require wrapping the method so that it is compatible with the framework. (Additionally, Bai et al. [2024] noticed that the PACS dataset is quite a bit different from other DG datasets so seeing on other datasets in the benchmark would be helpful.)

- The explanation of the method and the key insights are not written well. Several parts are simply incorrect though I can almost guess at what is meant. Others may be correct but are not explained well or justified appropriately. Please see questions below.

[Bai et al., 2024] Bai, R., Bagchi, S., & Inouye, D. I. (2024). Benchmarking algorithms for federated domain generalization. ICLR.

**Questions:**

- Why do you formulate this as a bi-level optimization problem? Why is this necessary? It is important to lead the reader up to this point rather than just stating it. Furthermore, this is not even a bi-level problem in 3a and 3b. This is just 2 equations. Are you minimiizing 3a subject to 3b? If so, it's not even clear that 3b is minimization problem, it's just a constraint perhaps? This is incorrectly formalized.

- What is the intuition between the difference between yours and [Shi et al., 2022]? Is it something like the some of inner pairwise products is bounded by the sum of inner products between a mean vector and each vector? Does this have relationship to standard sum of squares ideas?

- How is the update rule for your method different than FedAvg? It seems like a different aggregation method but is difficult to understand how it is different than simple FedAvg. Could you provide an more explicit comparison and discussion on how it differs? When and why would the weights be different than FedAvg and when would the be equivalent?




- Other than computational, is there a reason to limit the search space to a convex combination of local gradients? This doesn't seem necessary.

- Why is a search space limitation needed? This is not well-motivated but just stated as fact. It seems that constraining to a convex combination + constraining to be near a simpler method like FedAvg strongly regularizes the method. But it is not clear why this is necessary or justified, perhaps other than an empirical argument.

- Eq 11 is not a multi-objective optimization problem as it is written. Essentially it has been reduced to a single objective via scalarization $\gamma$ parameter. Thus, it is not clear why multi-objective optimization is needed or required.

- Eq 11 - Why is $\kappa$ needed? It seems that this term does not depend on $\Gamma$ and thus can be ignored.

- There is no lead up to Theorem 1 and this theorem is not well-explained. Why is this an easier problem? Why did you need to go through the min-max problem setup? This whole derivation seems very convoluted and does not lead logically to the next step. A major revision and explanation is needed.

- The theoretic analysis seems to use the same basic tools and techniques from prior works. Could you briefly explain the similarity of each main lemma, theorem or corollary w.r.t. its prior (probably non-FL) counterparts? Are there any new theoretic techniques used?

*Summary of Review*
The basic idea of doing gradient matching on the server seems natural and the paper proposes one practical way to implement this. Furthermore, the empirical results are fairly strong either by itself or in combination with other methods. However, the method is not explained well and the theoretic analysis seems very similar to prior work (useful but not itself much of a contribution).  The lack of comparison to a Federated DG benchmark paper from last year's ICLR also calls the results into question.

---

> ### Author Response · Authors · 2024-11-19
> **Response to Reviewer XXzG's Weaknesses**
>
> > **Weakness 1**: The paper does not compare to the federated domain generalization benchmark [R1] that exactly fits this setting. This benchmark paper enables comparison of your method to multiple prior DG methods adapted from central methods and federated DG methods. In particular, [R1] noticed that simple federated averaging or convergence-specific FL methods could beat most prior Federated DG methods. [R1] also control for hyperparameter tuning costs. I would like to see this method applied within this framework to compare to the methods in that benchmark. This will place FedOMG on the same playing field as prior methods and would only require wrapping the method so that it is compatible with the framework. (Additionally, [R1] noticed that the PACS dataset is quite a bit different from other DG datasets so seeing on other datasets in the benchmark would be helpful.)
>
> In our work, we compare our algorithm with recent SOTA algorithms in FDG, i.e., FedGA, FedSR, FedSAM, StableFDG. They are already peer-reviewed in flagship conferences. We inherited the experimental settings and benchmarking from FedGA, FedSAM official repository, which we believe the evaluation is comprehensive and widely approved.
>
> Based on our discussions regarding DG algorithms, it appears that their suitability for federated settings is limited due to the requirement for data accessibility among devices or domains. As such, we believe that direct comparisons between FDG and DG algorithms may not be entirely appropriate in this context.
>
> We agree that various datasets can bring different characteristics. Besides PACS, we also evaluate our FDG on VLCS and OfficeHome. We are considering suggested [R1] and going on two more datasets (i.e., IWildCAM and CelebA).
> The actual results of the benchmark can be found in the anonymous wandb link as follows: https://wandb.ai/anonymous12/FL_DG_Benchmark
>
> > **Weakness 2**: The explanation of the method and the key insights are not written well. Several parts are simply incorrect though I can almost guess at what is meant. Others may be correct but are not explained well or justified appropriately. Please see questions below.
>
> We thank the Reviewer for the constructive comments. We have majorly revised the paper, especially the notations to make the paper more consistent and reduce some issues, the major revision including.
> - We give more explanation to some derivations (e.g., Theorem 2) to make the paper more accessible.
> - According to question $1$, we have revised Section 3.1 majorly to improve the paper clarity and make the story more connected.
> - According to question $2$, we have added and highlight the discussion about the difference between FedOMG and Fish, which also explain the significance of our works compared to SOTA of vanilla domain generalization. We respectfully acknowledge that the Reviewer has pointed out one of our contributions that we have missed during the paper finalization. However, due to the shortage of rebuttal time and time to evaluate the algorithm on new benchmark, we will add the revised version with additional results in the last days of the rebuttal phase.
> - According to question 4-5-6, we have revised and provided more explanation to our revised manuscript to improve the paper clarity and also improve the contribution and significance of our paper.
> - According to question $8$, we have provided a more detailed explanation in our revised manuscript to improve the clarity of the lead up to Theorem 1.

---

> > ### Comment · Reviewer_XXzG · 2024-11-21
> > **Wrong metric for considering Fed DG benchmark**
> >
> > Hi authors,
> >
> > While I will read and respond the rest of the response soon, I wanted to quickly note that you are not looking at the right metric for the Federated DG benchmark paper. For CelebA, you should be looking at the "Worst Group Accuracy" rather than the "Average Accuracy" when comparing to results in the paper. See https://github.com/inouye-lab/FedDG_Benchmark/blob/main/src/dataset_bundle.py#L355 showing that metric is "acc_wg", where "wg" refers to "Worst Group". Each dataset has a specific application-specific metric defined by the original WILDS dataset and depending on the scenario.
> >
> > This calls into question your precision and ability to run careful experiments as you have made a claim that a previous paper is wrong without carefully understanding the experimental setup. If something is surprising, you should strongly question whether you did something incorrect and making entirely certain that you are correct before making claims.

---

> ### Author Response · Authors · 2024-11-19
> **Response to Reviewer XXzG's Questions Part 1**
>
> >**Question 1**: Why do you formulate this as a bi-level optimization problem? Why is this necessary? It is important to lead the reader up to this point rather than just stating it. Furthermore, this is not even a bi-level problem in 3a and 3b. This is just 2 equations. Are you minimizing 3a subject to 3b? If so, it's not even clear that 3b is minimization problem, it's just a constraint perhaps? This is incorrectly formalized.
>
> We apologize for the unclear description of bi-level optimization problem and have substantially revised the manuscript to improve clarity.
> According to the reason of formulating Eq. (3), our approach is to utilize meta-learning principles to decompose the joint learning function of FDG in Eq. (2), into two learning steps: a local update and a meta update.
> - In the first steps, local update in Eq. (3b) is applied on the client side.
> - In the second steps, meta update in Eq. (3a) is applied on the server side.
>
> The purpose of disentanglement is to enable an approach to designing the on-server optimization process based on Equation (3a). Specifically, the on-server optimization is derived from Equation (3a) and formalized in the resulting Theorem 1.
>
> >**Question 2**: What is the intuition between the difference between yours and $[\textrm{R1}]$? Is it something like the sum of inner pairwise products are bounded by the sum of inner products between a mean vector and each vector? Does this have relationship to standard sum of squares ideas?
>
> Fish algorithm $[\textrm{R1}]$ proposes the utilization of inner products to consider the gradients among domains. However, Fish is not suitable in Federated settings for two following issues.
> - As mentioned by the authors of Fish, the direct utilization of inner products may induce a huge computation overhead (as they require the second-order derivative of the model’s parameters due to the gradient inner product term (see Section 3.4 of $[\textrm{R1}]$).
> - To deal with the first issue, Fish’s authors introduce an indirect applying continuously model update like Reptile. This approach is infeasible in Federated settings, as the models are required to be transmitted among clients continuously, thus, inducing significant communication overheads.
>
> Our FedOMG overcomes these two challenges. Meanwhile, we have provided a discussion about the difference between our FedOMG and Fish in the revised manuscript.
>
> - $[\textrm{R1}]$ [Shi et al., 2022] Gradient Matching for Domain Generalization.
>
> >**Question 3**: How is the update rule for your method different than FedAvg? It seems like a different aggregation method but is difficult to understand how it is different than simple FedAvg. Could you provide a more explicit comparison and discussion on how it differs? When and why would the weights be different than FedAvg and when would then be equivalent?
>
> To discuss about the difference between FedOMG and FedAvg, we first revisit the gradient update (each communication round $r$) as follows:\
> $$\theta^{(r+1)}\_g = \theta^{(r)}\_g - \sum^{U}\_{u=1}\gamma_u g^{(r)}\_u,$$
> where we optimize the variable $\gamma\_u$ via Eq. 12b. The key differences between FedAvg and FedOMG are that
> - In FedAvg, the optimization variable set $\Gamma = \lbrace \gamma\_u\vert\forall u\in \mathcal{U}\_\mathcal{S} \rbrace $ is distributed uniformly, i.e., $\gamma\_u = \gamma\_v, \forall u,v \in \mathcal{U}\_\mathcal{S}$. This approach is proven to induce the weight divergence $[R1]$.
> - In FedOMG, the optimization variable set $\Gamma = \lbrace \gamma_u\vert\forall u\in\mathcal{U}_\mathcal{S} \rbrace $ is not distributed uniformly. Most notably, $\Gamma$ can be optimized such that the aggregated gradient can achieve invariant gradient direction characteristics by maximizing the inner product between aggregated gradient and clients' gradients from Eq. (12b), i.e., $\Gamma\_\textrm{IGD} = \arg max\_{\Gamma} \sum\_{u\in \mathcal{U}\_\mathcal{S}} \Big\langle\Gamma \mathbf{g}^{(r)},g^{(r)}\_u\Big\rangle$.
> - $[\textrm{R1}]$ [Zhao et al., 2018] Federated Learning with Non-IID Data.

---

> ### Author Response · Authors · 2024-11-19
> **Response to Reviewer XXzG's Questions Part 2**
>
> >**Question 4-5-6**: Other than computational, is there a reason to limit the search space to a convex combination of local gradients? This doesn't seem necessary. \
> Why is a search space limitation needed? This is not well-motivated but just stated as fact. It seems that constraining to a convex combination + constraining to be near a simpler method like FedAvg strongly regularizes the method. But it is not clear why this is necessary or justified, perhaps other than an empirical argument. \
> Eq 11 - Why is $\kappa$ needed? It seems that this term does not depend on $\Gamma$ and thus can be ignored.
>
> We apologize for the lack of clarity in our manuscript. Based on the Reviewer’s feedback, we have now provided a more comprehensive and precise explanation in the revised manuscript. \
> As mentioned in our paper, we use a global gradient of FL baseline (e.g., FedAvg) as a reference. From the given reference, our objective is to find an optimal gradient which achieve the invariant gradient direction. From this point, when the searching space too large or no searching space limitation is applied, there are two challenges occurs:
> - The on-server optimization requires more iterations to converge the optimal results. Therefore, FedOMG induces more computation overheads.
> - As the optimization problem focuses on maximizing the GIP among users $\langle\Gamma \mathbf{g}^{(r)}, g^{(r)}\_u\rangle$, the optimization may lead to the optimization bias toward the gradients with most dominating gradient magnitude. As a consequence, the optimization of GIP will lose the generalization and may forget the clients which does not contribute much that communication round. By applying the limited searching space, we can limit the searched gradient not being bias too far from the reference, thus retaining the generalization capability of the FL algorithm used as a reference.
> -  An alternative approach for GIP is using cosine similarity. However, gradient norm at the denominator of cosine similarity has high computation overheads, and thus, cosine similarity is also infeasible to relax to a more simplified version as in Theorem 1.
>
> All in all, we believe that limiting the searching space is necessary for FedOMG. However, we acknowledge that our lack of explanation induces the confusion. Thus, we have revised and provided more explanation to our revised manuscript to improve the paper clarity and also improve the contribution and significance of our paper.
>
> The parameter $\kappa$ is needed in the equation because it is used to control the radius of searching space and operates as a hyper-parameter of FedOMG. As explained above, we believe that $\kappa$ is also crucial to the generalization of our proposed FedOMG.
>
> >**Question 7**: Eq. 11 is not a multi-objective optimization problem as it is written. Essentially it has been reduced to a single objective via scalarization $\gamma$ parameter. Thus, it is not clear why multi-objective optimization is needed or required.
>
> We apologize for the confusion about the multi-objective term in Eq. (11). We have removed the sentence in the revised paper as the sentence is redundant to improve the clarity.
>
> >**Question 8**: There is no lead up to Theorem 1 and this theorem is not well-explained. Why is this an easier problem? Why did you need to go through the min-max problem setup? This whole derivation seems very convoluted and does not lead logically to the next step. A major revision and explanation are needed.
>
> Min-max problem in Eq. (12) arises from applying Lemma 2 to Eq. (11). The introduction of the min-max problem is to formulate Theorem 1, as the min-max problem is solvable through convex optimization (see Appendix E.5). This approach simplifies the process by reducing the need to compute the argmax over a summation of $U$ clients in Eq. (11). As a consequence, we can design an argmin function involving only two vectors in Eq. (13).

---

> ### Author Response · Authors · 2024-11-19
> **Response to Reviewer XXzG's Questions Part 3**
>
> >**Question 9**: The theoretic analysis seems to use the same basic tools and techniques from prior works. Could you briefly explain the similarity of each main lemma, theorem or corollary w.r.t. its prior (probably non-FL) counterparts? Are there any new theoretic techniques used?
>
> The new theoretic techniques in our paper lie in two terms:
> - In the Lemma 3, we measure the domain divergence by the expectation of the gradient divergence. It is noteworthy that the current works consider the domain divergence with the estimated loss between two domains $[\textrm{R1}]$, or did not take the domain divergence into consideration $[\textrm{R2}]$. By proposing Lemma 3 and applying it to derive Theorem 2, we prove that by minimizing the gradient divergence $\sum\_{v\in \mathcal{U}\_\mathcal{S}} \frac{d\_{\mathcal{G}\circ\theta}(\hat{\mathcal{D}}_u, \hat{\mathcal{D}}_v)}{\mu} $ the our generalization gap can be significantly reduced in comparison with current works.
> - In Theorem 2, we apply the disentanglement on the domain shift between the source and target dataset $d_{\mathcal{H}\bigtriangleup\\mathcal{H}}(\mathcal{D}\_\mathcal{S}, \mathcal{D}\_\mathcal{T})$ (which is proven by $[\textrm{R1}]$) into $\sum\_{v\in \mathcal{U}\_\mathcal{S}} \frac{d\_{\mathcal{G}\circ\theta}(\hat{\mathcal{D}}_u, \hat{\mathcal{D}}_v)}{\mu} + d\_{\mathcal{H}\bigtriangleup\\mathcal{H}}(\mathcal{D}\_\mathcal{S}, \mathcal{D}\_\mathcal{T}) $.
>
> The disentanglement process proves advantageous by decomposing domain divergence into two components: a reducible term, $\sum\_{v\in \mathcal{U}\_\mathcal{S}} \frac{d\_{\mathcal{G}\circ\theta}(\hat{\mathcal{D}}_u, \hat{\mathcal{D}}_v)}{\mu} $, and an irreducible term, $d\_{\mathcal{H}\bigtriangleup\\mathcal{H}}(\mathcal{D}\_\mathcal{S}, \mathcal{D}\_\mathcal{T})$ remains non-reducible due to the inaccessibility of the target dataset $\mathcal{D}\_\mathcal{T}$ resulting in a persistent divergence. This disentanglement introduces new perspectives in Federated Domain Generalization by enabling a focused effort to minimize the reducible term, $\sum\_{v\in \mathcal{U}\_\mathcal{S}} \frac{d\_{\mathcal{G}\circ\theta}(\hat{\mathcal{D}}_u, \hat{\mathcal{D}}_v)}{\mu} $ which could significantly enhance the generalization capability of FDG systems.
>
> - $[\textrm{R1}]$ Ruipeng Zhang et al., Federated Domain Generalization with Generalization Adjustment, CVPR 2023.
> - $[\textrm{R2}]$ A. Tuan Nguyen et al., FedSR: A Simple and Effective Domain Generalization Method for Federated Learning, NIPS 2022.

---

> ### Author Response · Authors · 2024-11-21
> **Any follow-up question?**
>
> Dear Reviewer XXzG,
>
> We sincerely appreciate your efforts and time for the community. As we approach the close of the author-reviewer discussion period in one week, we wonder whether the reviewer is satisfied with our response. It will be pleasurable if the reviewer can give us the reviewer's thoughts on the current revision to give us an extra valuable chance to improve our paper. We summarized our revision in the "Revision summary" comment.
>
> Again, we thank the reviewer's valuable commitment and their help to strengthen our submission. We will address all the raised concerns by reviewers if there remain any.

---

> > ### Comment · Reviewer_XXzG · 2024-11-21
> >
> > Hi authors, I have read through your response and update. I appreciated your attempts to clarify the method. I think these changes have improved the manuscript's clarity.  The weakness of not comparing to the federated DG benchmark paper is still concerning along with the misunderstanding of the experiments in the benchmark paper (see other comment) only increases my concern. I do not have any further questions.

---

> > > ### Author Response · Authors · 2024-11-24
> > > **About the Experimental Evaluations on FL-DG Benchmark**
> > >
> > > Dear Reviewer XXzG,
> > >
> > > Thank you for your clarification regarding the evaluation metric for the Celeb-A dataset. Following your suggestion, we have
> > > evaluated our results on the Celeb-A dataset and added the results into the revised manuscript.
> > >
> > > Additionally, we have included relevant references, including those pertaining to benchmarks and the FedADG algorithm. We welcome any further feedback or suggestions from you on how we might further improve our manuscript.
> > >
> > > Sincerely,
> > >
> > > Authors

---

> > > > ### Comment · Reviewer_XXzG · 2024-12-02
> > > >
> > > > Hi authors,
> > > >
> > > > I appreciate the new results on Celeb-A with the Fed DG benchmark. This gives more confidence in the approach. I would recommend integrating these results and more difficult datasets from the Fed DG benchmark to give stronger evidence of your approach in the final manuscript if accepted.
> > > >
> > > > Also, please remove your incorrect statements about the benchmark due to your misunderstanding of the benchmark metrics given the public nature of these comments. You should edit/retract your comments that are incorrect as soon as you notice them. I'm quite surprised you have left them unedited even now.

---

> ### Author Response · Authors · 2024-12-02
>
> Dear Reviewer XXzG,
>
> We sincerely appreciate the Reviewer's constructive comments and will revise accordingly. Due to time constraints, we were only able to conduct evaluations on the CelebA dataset. However, we are currently running experiments on more challenging datasets and will include these results.
>
> We can also provide the integration of the code into FedDG benchmark in the future.
>
> Sincerely,
>
> Authors

---

### Author Response · Authors · 2024-11-19
**Revision uploaded**

We thank all the reviewer for their comments, and we have published the updated manuscript with the following major changes:
1. [XXzG, cQAL] Provide more details about the paper motivations and contributions (Section 1)
2. [XXzG] Revised and provide explanation of on-server optimization (Section 3)
3. [XXzG] Provided the discussion about the difference between FedOMG and Fish (Section 3)
4. [XXzG] Provided more explanation for the reason of using limiting searching space (Section 4)
5. [XXzG] Provided a more detailed explanation in our revised manuscript to improve the clarity of the lead up to Theorem 1 (Section 4)
6. [9XVR] Provided more detailed explanation about motivations of Invariant Gradient Direction (Section 3)
7. [9XVR] Provided explanation about the estimated computation (Section 3)
8. [HBia] Provided Hyper-parameters for FedOMG to prove that FedOMG is computation efficient on the server side (Appendix D.5)
9. [cQAL] Revised to improve clarity and notation consistency (Section 4, 5, Appendix F.4).
10. [cQAL] Provided more explanations about the Indirect Search of Invariant Gradient Direction (Section 4).

We are also considering the benchmark according to Reviewer XXzG and the Illustrative Toy Task according to Reviewer cQAL. The revised manuscript may be further updated in the future.

**Updated on 23-11-2024**: Per the Reviewer XXzG suggestion and clarification, we have provided the additional results on Celeb-A dataset with two non-IID settings ($\alpha = 0.1, 1.0$). The results are provided in Appendix E.5.

---

### Public Comment · ~Minh-Duong_Nguyen1 · 2025-03-02
**Revision Updates**

Dear PCs, SACs, ACs, and Reviewers,

In our camera-ready version, we have updated the title from "Federated Domain Generalization with Data-free On-server Gradient Matching" to "Federated Domain Generalization with Data-free On-server Matching Gradient." This change was made to better align with the proposed method and the abbreviated term introduced in our paper. Since "Gradient Matching" and "Matching Gradient" convey the same meaning, we believe this revision does not significantly alter the original title.

Sincerely,

Authors

---

### Meta-Review · Area_Chair_yfXt · 2024-12-26

**Metareview:**

The paper introduces FedOMG, a method aimed at improving Federated Domain Generalization (FDG) by leveraging a gradient-matching strategy. This approach uses a meta-learning framework to find an optimal combination of local updates, which is more efficient than the simple averaging used in traditional methods like FedAvg. The paper presents both theoretical results and empirical evidence demonstrating that FedOMG significantly improves performance over baseline methods across several datasets, and that it can be combined with existing Federated Learning (FL) methods to boost performance further. Most of the concerns raised by reviewers are (partially) revised by authors and some are already incorporated in the revised version and the consensus is on accepting the paper.

**Additional Comments On Reviewer Discussion:**

There are several areas where the paper could be improved. One key limitation is the lack of comparison with a recent Federated Domain Generalization benchmark, particularly the work by Bai et al. (2024), which could place FedOMG in the context of existing methods and provide a clearer understanding of its strengths and weaknesses. Without this comparison, it is hard to assess whether the method is truly competitive with other state-of-the-art approaches. Another issue is the clarity of the paper’s presentation. Several sections, particularly around the motivation for the proposed approach, are not explained well. Additionally, the complexity of the optimization methods employed, including convex optimization and Pareto optimality, raises questions. The paper would benefit from a more intuitive explanation of how the gradient alignment contributes to domain invariance and why these complex techniques are necessary.

---

### Decision · Program_Chairs · 2025-01-22

Accept (Poster)